# The Secondary Formation of Organosulfates under the Interactions between Biogenic Emissions and Anthropogenic Pollutants in Summer of Beijing

Yujue Wang,[1] Min Hu,[*,1,5] Song Guo,[1] Yuchen Wang,[3] Jing Zheng,[1] Yudong Yang,[1] Wenfei Zhu,[6] Rongzhi Tang,[1] Xiao Li,[1] Ying Liu,[1,5] Michael Le Breton,[2] Zhuofei Du,[1] Dongjie Shang,[1] Yusheng Wu,[1] Zhijun Wu,[1] Yu Song,[1] Shengrong Lou,[6] Mattias Hallquist,[2] and Jianzhen Yu [*,3,4]

[1]State Key Joint Laboratory of Environmental Simulation and Pollution Control, College of Environmental Sciences and Engineering, Peking University, Beijing 100871, China

[2]Department of Chemistry and Molecular Biology, University of Gothenburg, Gothenburg, Sweden

[3]Environmental Science Programs, Hong Kong University of Science & Technology, Hong Kong, China

[4]Department of Chemistry, Hong Kong University of Science & Technology, Hong Kong, China

[5]Beijing Innovation Center for Engineering Sciences and Advanced Technology, Peking University, Beijing 100871, China

[6]Shanghai Academy of Environmental Sciences, Shanghai 200233, China

*Correspondence to*: Min Hu (minhu@pku.edu.cn); Jianzhen Yu (jian.yu@ust.hk)

**Abstract.** Organosulfates (OSs), with ambiguous formation mechanisms, are a potential source of "missing secondary organic aerosol (SOA)" in current atmospheric models. In this study, we chemically characterized OSs and nitrooxy OSs (NOSs) formed under the influence of biogenic emissions and anthropogenic pollutants (e.g. $NO_x$, $SO_4^{2-}$) in summer of Beijing. An ultrahigh-resolution mass spectrometer equipped with electrospray ionization source was applied to examine the overall molecular composition of S-containing organics. The number and intensities of S-containing organics, the majority of which could be assigned as OSs and NOSs, increased significantly during pollution episodes, which indicated their importance for SOA accumulation. To further investigate the distribution and formation of OSs and NOSs, high performance liquid chromatography coupled to mass spectrometry was employed to quantify ten OSs and three NOS species. The total concentrations of quantified OSs and NOSs were 41.4 and 13.8 ng/m$^3$, respectively. Glycolic acid sulfate was the most abundant species among all the quantified species, followed by monoterpene NOSs ($C_{10}H_{16}NO_7S^-$). The total concentration of three isoprene OSs was 14.8 ng/m$^3$ and the isoprene OSs formed via $HO_2$ channel was higher than those formed via $NO/NO_2$ channel. The OS concentration coincided with the increase of acidic sulfate aerosols, aerosol acidity and liquid

water content (LWC), indicating the acid-catalyzed aqueous-phase formation of OSs in the presence of acidic sulfate aerosols. When sulfate dominated the accumulation of secondary inorganic aerosols (SIAs, sulfate, nitrate and ammonium) ($SO_4^{2-}$/SIAs> 0.5), OS formation would be obviously promoted as the increasing of acidic sulfate aerosols, aerosol LWC and acidity (pH< 2.8). Otherwise, the acid-catalyzed OS formation would be limited by lower aerosol acidity when nitrate dominated the SIA accumulation. The nighttime enhancement of monoterpene NOSs suggested their formation via nighttime $NO_3$-initiated oxidation of monoterpene under high-$NO_x$ conditions. However, isoprene NOSs are supposed to form via acid-catalyzed chemistry or reactive uptake of oxidation products of isoprene. This study provides direct observational evidence and highlights the secondary formation of OSs and NOSs, via the interaction between biogenic precursors and anthropogenic pollutants ($NO_x$, $SO_2$ and $SO_4^{2-}$). The results imply that future reduction in anthropogenic emissions can help to reduce the biogenic SOA burden in Beijing or other areas impacted by both biogenic emissions and anthropogenic pollutants.

# 1    Introduction

Secondary organic aerosols (SOA), formed by atmospheric oxidation of volatile organic compounds (VOCs), accounts for a large fraction of organic aerosols (OA) on the global scale    (Jimenez et al., 2009; Guo et al., 2014). However, current models usually underestimate (Kroll and Seinfeld, 2008; Hallquist et al., 2009) or predict the SOA concentration with large uncertainties (Jimenez et al., 2009; Kiehl, 2007; Shrivastava et al., 2017) in the ambient atmosphere. Thus, it is important to elucidate potential missing groups of compounds or formation mechanisms. Organosulfates (OSs), commonly formed via the interaction between VOC precursors and acidic sulfate seed particles, could be a potential source of "missing SOA" in current atmospheric models (Surratt et al., 2010). OSs have been observed in various ambient atmospheres, including urban, rural, suburban, forest as well as remote environments (Lin et al., 2012; Meade et al., 2016; Stone et al., 2012; Riva et al., 2015; Brüggemann et al., 2017), which could represent 2-30% of OA (Hawkins et al., 2010; Stone et al., 2012; Frossard et al., 2011; Tolocka and Turpin, 2012; Surratt et al., 2008; Liao et al., 2015).

Many prior chamber experiments revealed the precursors and formation mechanisms of OSs (Surratt et al., 2007;

Surratt et al., 2010; Surratt et al., 2008; Liggio and Li, 2006; Chan et al., 2011; Shalamzari et al., 2014; Shalamzari et al., 2016; Zhang et al., 2012), however, the atmospheric relevance of these remain unclear. Various biogenic VOCs (BVOCs) precursors have been reported, including isoprene (Hatch et al., 2011; Surratt et al., 2010), monoterpenes (Surratt et al., 2008), sesquiterpenes (Chan et al., 2011), pinonaldehyde (Liggio and Li, 2006), unsaturated aldehydes (Shalamzari et al., 2014; Shalamzari et al., 2016) and 2-methyl-3-buten-2-ol (Zhang et al., 2012). OSs originating from isoprene are some of the most studied compounds and could be among the most abundant OA in some areas (Liao et al., 2015; Chan et al., 2010; Surratt et al., 2010; Lin et al., 2013a; Worton et al., 2013). Isoprene OSs usually form through ring-opening epoxide chemistry catalyzed by acidic sulfate aerosols (Worton et al., 2013; Froyd et al., 2010; Paulot et al., 2009). OSs were also proposed to form by reactive uptake of VOCs or their oxidation products that involves the sulfate radicals (Nozière et al., 2010; Schindelka et al., 2013). The sulfate esterification of alcohols could also be a pathway leading to OSs formation, while Minerath et al (2018) predicted that this mechanism was kinetically insignificant under ambient tropospheric conditions. However, this prediction was based on laboratory bulk solution-phase experiments and the applicability to the liquid-phase on particles suspended in the air is unconfirmed. Nitrooxy organosulfates (NOSs) were observed to form via the nighttime $NO_3$-initiated oxidation of VOC precursors (e.g. monoterpene), followed by alcohol sulfate esterification (Iinuma et al., 2007; Surratt et al., 2008). Organic nitrate ($R-ONO_2$) could also act as precursors to OSs through the nucleophilic substitution of nitrate by sulfate (Hu et al., 2011; Darer et al., 2011).

Both aerosol acidity and liquid water content (LWC) are key variables influencing the OS formation processes. OS formation could only happen in the presence of sulfate aerosols, enhanced by increased aerosol acidity, through acid-catalyzed reactive uptake and multiphase reactions of oxidation products (Riva et al., 2016c; Surratt et al., 2010; Lal et al., 2012; Riedel et al., 2015). Previous studies also demonstrated the importance of aqueous-phase or heterogeneous reactions for OS formation (Lal et al., 2012; McNeill et al., 2012; McNeill, 2015; Riedel et al., 2015). On one hand, the increased LWC would decrease the aerosol viscosity, which favors the exchange of organics or other gas molecules into the particles, mass diffusion of reactants and heterogeneous chemical reactions within the particles (Vaden et al., 2011; Booth et al., 2014; Renbaum-Wolff et al., 2013; Shrestha et al., 2015; Zhang et al., 2015), and thereby enhance the OS formation. On the other hand, more LWC would lead to increased pH due to dilution. For example, Riva et al. (2016) and Duporte et al.

(2016) found that the OS formation decreased with higher RH, which was attributed to the increased pH as a result of higher
LWC (Duporte et al., 2016; Riva et al., 2016c).
To get a comprehensive understanding of the characteristics and formation of OSs in the ambient atmosphere, it is
desirable to simultaneously identify and quantify particulate OSs on the molecular level. Soft ionization techniques coupled
with ultrahigh-resolution mass spectrometer (UHRMS) have been widely applied to identify various and numerous organics,
including OS species, in ambient aerosols or chamber studies (Lin et al., 2012; Blair et al., 2017; Tao et al., 2014; Wang et al.,
2016). UHRMS is a powerful analytical tool in gaining an overall characterization of OSs, however, the quantification
capability is limited without pre-separation. High performance liquid chromatography coupled to mass spectrometer
(HPLC-MS) is suitable for the separation and quantification of different OS compounds. However, one noted limitation is a
lack of commercially available authentic standards. As a result, surrogate standards are often used for quantification (He et
al., 2014; Riva et al., 2015; Zhang et al., 2012), which adds uncertainty to the concentrations (Wang et al., 2017d). Recently,
a few research groups quantified some OS species using synthetic authentic standards (e.g. hydroxyacetone sulfate, glycolic
acid sulfate, lactic acid sulfate, methyltetrol sulfate, aromatic OSs, α/β-pinene OS,Limonene OS and Limonaketone OS)
(Hettiyadura et al., 2017; Hettiyadura et al., 2015; Olson et al., 2011; Wang et al., 2017d; Ma et al., 2014; Budisulistiorini et
al., 2015; Staudt et al., 2014), which was very important for understanding the variation and formation of OSs in ambient
aerosols.
Missing knowledge of formation mechanisms, the complexities of ambient aerosol composition and oxidation condition,
and the lack of commercially available standards all hinder us from understanding the formation and fate of OSs in ambient
atmosphere. Few field studies has been conducted in urban areas dominated by anthropogenic pollutants (e.g. $NO_x$, $SO_4^{2-}$).
Observations are lacking to illustrate how severe anthropogenic pollutants could influence the OS formation under different
physical environmental conditions. This work reports a comprehensive characterization of particulate OSs in summertime
Beijing, a location under the influence of both biogenic and severe anthropogenic sources. This study provides direct
observational evidence for gaining insights into OS formation. Orbitrap MS coupled with soft ionization source was used to
identify the overall molecular composition of S-containing organics. HPLC-MS was then applied to quantify some OSs and
NOS species in ambient aerosols using newly synthesized authentic standards and surrogate standards. Previously proposed
formation pathways of OS or NOS (e.g. acid-catalyzed aqueous-phase chemistry, nighttime $NO_3$ chemistry) were considered,
and the influence of different environmental conditions or factors on the formation were comprehensively elaborated. It has
been suggested that both aqueous-phase chemistry and nighttime $NO_3$ chemistry play important roles in the heavy haze of
Beijing (Wu et al., 2018; Wang et al., 2017b; Wang et al., 2017a). Using OSs and NOSs as examples, this work illustrates
SOA formation via acid-catalyzed aqueous-phase chemistry, nighttime $NO_3$ chemistry under the interaction between
abundant anthropogenic pollutants and biogenic emissions.
**2    Methods**
**2.1 Sample collection**
This study was a part of the bilateral Sweden-China framework research program on 'Photochemical smog in China:
formation, transformation, impact and abatement strategies', focusing on the SOA formation under the influence of
anthropogenic pollutants (Hallquist et al., 2016). An intensive field campaign was conducted at Changping (40.14° N,
116.11° E), a regional site 38 km northeast of the Beijing urban area, China. The campaign was conducted from May 15 to
June 23, 2016, when the site was influenced by high biogenic emissions from vegetation in the nearby mountains and
anthropogenic pollutants from the nearby villages and Beijing urban areas (Tang et al., 2017). During May 17- June 5, the
average concentrations of isoprene, monoterpenes, benzene, toluene and $NO_x$ were 297, 83, 441, 619 pptv and 22.7 ppb,
respectively.
Ambient aerosols were collected from May 16 to June 5. $PM_{2.5}$ (particles with aerodynamic diameter less than 2.5 μm)
samples were collected on prebaked quartz fiber filters (Whatman Inc.) and Teflon filters (Whatman Inc.) using a
high-volume sampler (TH-1000C, Tianhong, China) and a 4-channel sampler (TH-16A, Tianhong, China), respectively. The
sampling flow rates were 1.05 $m^3$/min and 16.7 L/min, respectively. The daytime samples were collected from 8:30 to 17:30
and nighttime ones from 18:00 to 8:00 the next morning. Field blank samples were collected by placing filters in the
samplers with the pump off for 30 min. The period May 20 - June 3 will be discussed in this study.

**2.2 Orbitrap MS analysis**

An Exactive Plus-Orbitrap MS (Thermo Scientific Inc., Bremen, Germany) equipped with a heated electrospray ionization (ESI) source was used to identify the overall molecular composition of OSs. Details of the extraction and data analysis have been described in Wang et al. (2017c). Briefly, a portion of filter was extracted with ultrapure water in an ultrasonic bath for 40 min and the extracts were filtered with 0.45 μm pore size PTFE syringe filter (Gelman Sciences). The filter portion size was adjusted to yield ~200 μg OC in each extract, in order to decrease the variation of ion suppression arising from varying coexisting organic components. The influence of ion suppress was illustrated in the Appendix S1. The extract sample was then loaded onto a solid phase extraction (SPE) cartridge (DSC-18, Sigma-Aldrich, USA) to remove inorganic ions and low molecular weight (MW) organic acids (Lin et al., 2010), followed by elution with methanol. The methanol eluate was dried under a gentle stream of $N_2$ and re-dissolved in acetonitrile/water (1:1) solvent for Orbitrap MS analysis. Some selected OS species of low MW (e.g., isoprene-derived OSs such as $C_4H_7O_7S^-$, $C_5H_7O_7S^-$, and $C_5H_{11}O_7S^-$) would be removed by the SPE clean-up procedure and thus not detected by the direct infusion Orbitrap MS analysis (see section 3.1). We note that these OS species were detected by HPLC-MS in the sample extracts to which no SPE pretreatment procedure was applied (see section 2.3). This phenomenon was also reported in previous studies (Gao et al., 2006; Surratt et al., 2007).

The Orbitrap MS was operated in negative mode (ESI-). The mass calibration was conducted using a standard mixture of N-butylamine, caffeine, MAFA, sodium dodecyl sulfate, sodium taurocholate and Ultramark 1621, with the scan range set to be 90-900 m/z. The Orbitrap MS had a mass resolving power of 140,000 at $m/z$ =200. Each sample was analyzed for three times with at least 100 full-scan spectra acquired in each analysis. The recorded mass spectra were processed and exported using the Xcalibur software (V2.2, Thermo Scientific). Peaks with a signal-to-noise ratio ≥10 were exported. All the mathematically possible formulas for each ion were calculated with a mass tolerance of 2 ppm. Each exported molecular formula was allowed containing certain elements and limited by several conservative rules (Wang et al., 2017c). Elements $^{12}C$, $^1H$, $^{16}O$, $^{14}N$, $^{32}S$ and $^{13}C$ were allowed in the molecular formula calculations. The H/C, O/C, N/C and S/C ratios were limited to 0.3- 3.0, 0- 3.0, 0- 0.5 and 0- 2.0. The assigned formulas were also restrained by the double bond equivalent values

and the nitrogen rule for even electron ions. More details about the molecular formula assignment have been introduced in
Wang et al. (2017c). The background spectra were obtained by analyzing the corresponding field blank sample following the
same procedure. Peaks were eliminated from the list if their intensities were lower than ten times of those in the blank
sample.
**2.3 Quantification of OSs and NOSs using HPLC-MS**
An aliquot of 25 cm$^2$ was removed from each filter sample and extracted in ultrasonic bath three times using 3, 2 and 1
mL methanol consecutively, each time for 30 min. The extracts were then filtered through a 0.25 μm polytetrafluoroethylene
(PTFE) syringe filter (Pall Life Sciences), combined, evaporated to dryness under a gentle stream of high-purity nitrogen and
re-dissolved in 50 μL methanol/water (1:1) containing 1 ppm D$_{17}$-octyl sulfate as internal standard. The solution was
centrifuged and the supernatant was used for analysis, using Agilent 1260 LC system (Palo Alto, CA) coupled to QTRAP
4500 (AB Sciex, Toronto, Ontario, Canada) mass spectrometer. The LC/MS was equipped with an ESI source operated in
negative mode. The optimized MS conditions and details of the method have been described in our previous study (Wang et
al., 2017d). Chromatographic separation was performed on an Acquity UPLC HSS T3 column (2.1 mm×100 mm, 1.8 μm
particle size; Waters, USA) with a guard column (HSS T3, 1.8 μm). The mobile eluents were (A) water containing 0.1%
acetic acid (v/v) and (B) methanol (v/v) containing 0.1% acetic acid at a flow rate of 0.19 mL/min. The gradient elution was
set as follows: the composition started with 1% B for 2.7 min; increased to 54% B within 12.5 min and held for 1.0 min; then
increased to 90% B within 7.5 min and held for 0.2 min; and finally decreased to 1% B within 1.8 min and held for 17.3 min
until the column was equilibrated. The column temperature was kept at 45 $^o$C and the injection volume was 5.0 μL.
The quantified OS and NOS species are listed in Table 1. The monoterpene NOSs (C$_{10}$H$_{16}$NO$_7$S$^-$ and C$_9$H$_{14}$NO$_8$S$^-$) were
quantified using the [M-H]$^-$ ions in the extracted ion chromatogram (EIC) and other species were quantified in
multiple-reaction monitoring (MRM) mode. OSs and NOSs were quantified using authentic standards or surrogates with
similar molecular structures (Table 1). Lactic acid sulfate (LAS) and glycolic acid sulfate (GAS) were prepared according to
Olson et al. (2011). The purity of LAS and GAS are 8% and 15%, determined by $^1$H NMR analysis using dicholoracetic acid
as an internal standard, and the recovery are 89.5% and 94.9%, respectively. Four monoterpene derived OS standards were
synthesized and the details are given in Wang et al. (2017). The purity of the four monoterpene OS standards are higher than
99% and the recovery are 80.5%-93.5% (Table S1). OSs with similar carbon chain structures usually have similar MS
responses (Wang et al., 2017d). Lactic acid sulfate was employed as a surrogate standard to quantify isoprene OSs due to
their similar structures and retention times (Table 1). α-pinene OS and limonaketone OS were respectively used to quantify
monoterpene NOSs $C_{10}H_{16}NO_7S^-$ and $C_9H_{14}NO_8S^-$ due to the similar carbon structures (Table 1). For the molecule with
isomers, quantification was performed by summing up the peak areas of the isomers, treated as one species (e.g.,
monoterpene NOSs with [M-H]$^-$ at $m/z$ 294 were treated as one NOS species).
**2.4 Other online and offline measurements**
A high resolution time-of-flight aerosol mass spectrometer (AMS) was employed to measure the chemical composition
of $PM_1$. The operation procedures and data analysis have been described in Zheng et al. (2017). VOCs were measured by a
proton-transfer-reaction mass spectrometer (PTR-MS). Meteorological parameters, including relative humidity (RH),
temperature, wind direction and wind speed (WS) were continuously monitored by a weather station (Met one Instrument
Inc.) during the campaign. Organic carbon (OC) was analyzed using thermal/optical carbon analyzer (Sunset Laboratory).
The organic matter (OM) concentration was calculated by multiplying OC by 1.6 (Turpin and Lim, 2001). Water soluble
inorganic ions and low MW organic acids (e.g. oxalic acid) were quantified by an ion chromatograph (IC, DIONEX,
ICS2500/ICS2000) following procedures described in Guo et al. (2010). After performing quality assurance/quality control
for IC measurements, the data (ions, pH, LWC) derived from IC measurements in the daytime samples of May 26 and 29
were excluded in the following analysis. Gaseous $NH_3$ was measured using a $NH_3$ analyzer (G2103, Picarro, California,
USA) (Huo et al., 2015). Aqueous phase [H$^+$] and LWC were then calculated with the ISORROPIA-II thermodynamic
model. ISORROPIA-II was operated in forward mode, assuming the particles are "metastable" (Hennigan et al., 2015;
Weber et al., 2016; Guo et al., 2015). The input parameters included: ambient RH, temperature, particle phase inorganic
species ($SO_4^{2-}$, $NO_3^-$, $Cl^-$, $NH_4^+$, $K^+$, $Na^+$, $Ca^{2+}$, $Mg^{2+}$), and gaseous $NH_3$. The thermodynamic calculations were validated by
the good agreement between measured and predicted gaseous $NH_3$ (slope=0.99, $R^2$= 0.97) (see Appendix S2 for details). The
contribution of organics to LWC was not considered in this study. Our previous study in Beijing has suggested that LWC
associated with organic species was insignificant (<6%), compared to that of secondary inorganic aerosols (Wu et al., 2018)
(see Fig. S3 for the comparison between LWC with or without water associated with organic compounds). Previous study
also suggested that the predicted aerosol acidity or pH without consideration of organic water could also be sufficient for
discussing aqueous SOA chemistry in this study, due to the minor effect on aerosol pH (0.15- 0.23) (Guo et al., 2015).
**3      Results and discussion**
**3.1 Overall molecular characterization of S-containing organics**
On average, 62% of the observed peaks in ESI negative mode are assigned with unambiguous molecular formulas. All
the assigned formulas were classified into four major categories based on their elemental compositions, including CHO,
CHON, CHOS and CHONS. As an example, CHONS refers to compounds that contain C, H, O, N and S elements in the
formula. Other compound categories are defined analogously. The percent of different compound categories in terms of
number and intensity are shown in Fig. S4 and Fig. 1, in which 'others' (e.g. CH, CHN, CHS, CHNS) refer to the
compounds excluded from the above major compound categories. During pollution episodes, the number and intensity
percent of S-containing compounds (CHOS and CHONS) increased obviously (Fig. 1, S4). The OC content in each sample
for Orbitrap MS analysis was kept roughly constant to minimize variation arising from matrix ion suppression. Taking the
nighttime sample of May 24 (0524N) as an example of clean days and the nighttime sample of May 30 (0530N) as an
example of polluted days, the mass spectra of different compound categories in each sample are shown and compared in Fig.
1 (a) and (b). The increase in S-containing organics indicated their important contribution to SOA when the pollution
accumulated. What's more, the S-containing compounds contributed more to the higher MW formulas than CHO ($O_1$-$O_{10}$) or
CHON ($O_1$-$O_{11}$) compounds (Fig. 1), due to the existence of more O (CHOS: $O_1$-$O_{12}$, CHONS: $O_1$-$O_{14}$) atoms and
heteroatoms (S, N) in the molecules. The increasing trend of S-containing organics (Fig S4), with larger MW than those of
CHO or CHON, may play important roles in the increase of SOA mass concentrations during pollution episodes.
The CHOS formulas with O/S≥ 4 allow the possible assignment of a sulfate group in the molecules (i.e., OSs) (Lin et
al., 2012). Among all the identified CHOS formulas, 60%-99% (93% on average) and 66-100% (96% on average) of them

could be assigned as OSs in terms of number and intensity percent. Analogously, the CHONS formulas with $O/(S+N) \geq 7$

could likely be NOSs formulas, which account for 22-78% (53% on average) by number and 18-94% (61% on average) by

intensity of all the identified CHONS formulas. As OSs and NOSs were assigned based on the molecular formulas alone, we

could not completely exclude the possibility of CHOS being hydroxysulfonates and CHONS being nitro-OSs due to the lack

of MS/MS analysis. According to previous study, the presences of organosulfonate or nitro-OSs were usually limited

compared to those of OSs or nitrooxy-OSs (Lin et al., 2012), thus they were not taken into consideration in this study. A total

of 351 OSs and 181 NOSs formulas were identified among all the samples during the campaign. The temporal variation of

the total number and intensity of OSs and NOSs are shown in Fig. S4. During pollution episodes (nighttime of May 27 - the

nighttime of May 28, nighttime of May 29 - the nighttime of May 30), the total number and intensity of OSs formulas

increased (Fig. S4). The total number of NOSs also showed similar increase trend during pollution episodes, while the total

intensity of NOSs showed nighttime enhancement during the whole observation period (Fig. S4). Previous studies suggested

that some NOS species could form via $NO_3$-initiated oxidation under high-$NO_x$ conditions at night (Surratt et al., 2008;

Iinuma et al., 2007; Gomez-Gonzalez et al., 2008), which will be further discussed in the following sections.

Some of the more abundant OSs and NOS peaks identified in the samples on the clean day (05/24N) or during pollution

episodes (05/30D, 05/30N) are listed in Table S2. For example, deprotonated molecules $C_9H_{15}SO_7^-$, $C_{10}H_{17}SO_7^-$ and

$C_9H_{17}SO_6^-$ were observed among the highest OS peaks in samples during pollution episodes (Table S2). These compounds

could be derived from the oxidation of alkanes or diesel fuel based on previous chamber studies (Riva et al., 2016c; Blair et

al., 2017). Many OSs previously designated as biogenic origins were also found in the anthropogenic sources (Blair et al.,

2017), which may raise uncertainty when assigning OS sources in field observation studies. OS compounds derived from

anthropogenic VOC precursors were widely observed in ambient aerosols (Table S2), while they were not quantified due to

the lack of standards in this paper. They will be further investigated in our future studies. Other OS molecules (e.g.

$C_9H_{15}SO_6^-$, $C_{10}H_{17}SO_5^-$) could be formed via the oxidation of monoterpenes (Surratt et al., 2008). For NOSs, ions

$C_{10}H_{16}NO_7S^-$, $C_{10}H_{16}NO_9S^-$ and $C_{10}H_{16}NO_{10}S^-$ were among the highest peaks (Table S2). They could form via the nighttime

$NO_3$-initiated oxidation of monoterpenes (Surratt et al., 2008). These are just some examples with higher relative intensity

(RI). The RI may not accurately represent their relative concentration levels in each sample, as the MS responses of different

OSs are also influenced by different carbon chain structures (Wang et al., 2017d). The OS species of low MW and short
carbon chain structures (with fewer than 6 carbon atoms in the molecule) are little retained on the SPE cartridges due to their
highly water-soluble and more hydrophilic properties (Gomez-Gonzalez et al., 2008; Lin et al., 2012; Lin et al., 2010). As
such, they were largely absent among the OS formulas detected by Orbitrap MS in this work. Hydroxyacetone sulfate
($C_3H_5O_5S^-$) was detected by Orbitrap MS only in several samples with relatively higher concentrations. Hydroxycarboxylic
acid sulfate ($C_2H_3O_6S^-$, $C_3H_5O_6S^-$) or isoprene OSs ($C_4H_7O_7S^-$, $C_5H_7O_7S^-$, $C_5H_{11}O_7S^-$) are also sufficiently hydrophilic that
little of them would be in the SPE eluate fraction, which was subjected for Orbitrap MS analysis. This explains why these
highly water-soluble OS species with lower MW are absent in Fig. 1. Though these OS species were not detected by Orbitrap
MS, some of them were quantified with high concentrations in the ambient aerosols in the LC/MS analysis (Table 1), as the
sample aliquots for the LC/MS analysis did not involve SPE treatment.

**3.2 Abundance of identified OSs and NOSs in ambient aerosols**

To further investigate the abundance and formation pathways of OSs and NOSs in ambient aerosols, some species were
then quantified by HPLC-MS using authentic standards when available or surrogate standards. The quantified species could
usually be formed via the interaction between biogenic precursors (e.g. isoprene, monoterpene) and anthropogenic pollutants
(e.g. $SO_4^{2-}$, $NO_x$), which have been reported in previous chamber studies (Surratt et al., 2007; Surratt et al., 2008; Surratt et
al., 2010). A total of ten OSs and three NOS species were quantified in this study and their concentrations are listed in Table
1. The molecules with the same molecular formula were treated as one species (e.g., monoterpene NOSs with [M-H]$^-$ at *m/z*
294 were treated as one NOS species). The average concentrations of all the quantified OSs were 41.4 ng/m$^3$ during the
campaign. The total OSs accounted for 0.31% of OM, with a maximum contribution of 0.65% on the night of May 30. The
total concentrations of quantified NOSs were 13.8 ng/m$^3$, corresponding to 0.11% of OM, with a maximum contribution of
0.35% on the night of May 23.
The concentrations of each OS or NOS species across this and prior studies were summarized in Table S3. The relative
contribution of each species to the total OSs or NOSs is shown in Fig. 2. GAS was the most abundant species among all the
quantified species. The concentrations of GAS were 3.9-58.2 ng/m$^3$, with an average of 19.5 ng/m$^3$. The concentrations were

higher than those observed in Mexico (4.1- 7.0 ng/m$^3$), California (3.3- 5.4 ng/m$^3$) or Pakistan (11.3 ng/m$^3$) (Olson et al.,

2011) (Table S3). The GAS concentration level at Beijing was comparable to those reported in summertime Alabama, US

(8-26.2 ng/m$^3$) (Table S3), a location characterized by high biogenic emissions and affected by anthropogenic pollutants

(Hettiyadura et al., 2015; Hettiyadura et al., 2017; Rattanavaraha et al., 2016). The concentrations of LAS were 0.7-12.0

ng/m$^3$, with an average of 4.4 ng/m$^3$. The LAS concentrations were also higher than those observed in Mexico (1.2-1.8

ng/m$^3$), California (0.6-0.8 ng/m$^3$) or Pakistan (3.8 ng/m$^3$), while lower than those observed in Alabama, US (16.5 ng/m$^3$)

(Olson et al., 2011; Hettiyadura et al., 2015; Hettiyadura et al., 2017) (Table S3). Carboxylic acids mainly form via

aqueous-phase oxidation in cloud or particle water, including both biogenic and anthropogenic sources (Charbouillot et al.,

2012; Chebbi and Carlier, 1996). The relatively higher level of hydroxycarboxylic acid sulfate could be attributed to the

favorable interaction between sulfate aerosols and carboxylic acids or other precursors in summertime Beijing, while the

precursors and mechanisms remain unclear. Oxalic acid is usually the most abundant dicarboxylic acid in the atmosphere

(Guo et al., 2010; Narukawa et al., 2003). The average concentration of oxalic acid in fine particles was 0.22 μg/m$^3$, which

was at a relatively high concentration level when comparing with those reported in previous studies (0.02-0.32μg/m$^3$)

(Agarwal et al., 2010; Bikkina et al., 2017; Boreddy et al., 2017; Deshmukh et al., 2017; Kawamura et al., 2010; Narukawa

et al., 2003). Strong inter-correlations were found among GAS, LAS and hydroxyacetone sulfate (HAS) (Table S4),

indicating their potentially similar precursors or formation pathways. They also showed strong correlations with isoprene

oxidation products (MVK+MACR) and isoprene OSs (Table S4), suggesting isoprene oxidized products as potential

precursors of GAS, LAS and HAS. It is suggested that both hydroxyacetone and carboxylic acids could be produced from

the oxidation of isoprene (Fu et al., 2008; Carlton et al., 2009). GAS, LAS and HAS have been reported to form via isoprene

oxidation in the presence of acidic sulfate (Riva et al., 2016b; Surratt et al., 2008). GAS was also observed to form via

sulfate induced oxidation of methyl vinyl ketone (MVK), oxidation product of isoprene (Schindelka et al., 2013).

The concentration of quantified isoprene OSs ($C_4H_7O_7S^-$, $C_5H_7O_7S^-$ and $C_5H_{11}O_7S^-$) was 14.8 ng/m$^3$, contributing to 36 %

of the total quantified OSs in this study. The isoprene OSs were lower than those observed in southeastern US, with

substantial isoprene emissions and impacted by anthropogenic pollutants, in which authentic standards were employed to

quantify the isoprene OSs (Rattanavaraha et al., 2016). We used lactic acid sulfate as a surrogate standard to quantify

isoprene OSs on the basis of their similar structures and retention times (Table 1). The isoprene concentration in southeastern US (1.9 ppb) (Xu et al., 2015) was much higher than that observed during our campaign (297 pptv). Besides the lower VOC precursors and measurement uncertainty, the lower isoprene OSs in this study could be attributed to different atmospheric conditions in Beijing from those in southeastern US. The IEPOX formation under low-$NO_x$ conditions ($HO_2$ channel), usually with higher yields than the oxidation products under high-$NO_x$ conditions ($NO/NO_2$) (Worton et al., 2013), could be suppressed under the high-$NO_x$ conditions (see section 3.4 for the high-$NO_x$ conditions) in Beijing (Zhang et al., 2017; Hu et al., 2015). The RH in Beijing was lower than that in southeast US (Xu et al., 2015), which possibly led to an increase of aerosol viscosity and a decrease of diffusivity within the particles, resulting in lower OS formation (Shiraiwa et al., 2011). Moreover, the OM-coated particle structures observed in Beijing could reduce the reactive uptake of isoprene oxidation products (Li et al., 2016; Zhang et al., 2018; Riva et al., 2016a), which may be another possible reason for lower isoprene OSs in this study. The concentrations were comparable to those observed in suburban area of mid-Atlantic or Belgium and higher than those observed at the background site of Pearl River Delta (PRD) region, China (Meade et al., 2016; Gómez-González et al., 2012; He et al., 2014), in which glycolic sulfate ester, ethanesulfonic acid or camphor sulfonic acid were employed as surrogate standards. The isoprene OSs formed via $HO_2$ channel ($C_5H_{11}O_7S^-$) were observed to be higher than that formed via $NO/NO_2$ channel ($C_4H_7O_7S^-$) (Table 1) (Worton et al., 2013). Isoprene had higher mixing ratio during the daytime (Fig. S5 (b)), when OH radicals dominated the atmospheric oxidation capacity. Furthermore, the yield of isoprene oxidation via $HO_2$ channel is proposed to be higher than that via $NO/NO_2$ channel (Worton et al., 2013). The concentration of $C_5H_7O_7S^-$ was comparable to that of $C_5H_{11}O_7S^-$ (Table 1). $C_5H_7O_7S^-$ was suggested to be formed via isoprene oxidation and related to $C_5H_{11}O_7S^-$ (Surratt et al., 2008), while the formation mechanism remains unclear. The concentration of isoprene NOSs ($C_5H_{10}NO_9S^-$) was lower than that of individual isoprene OSs. Strong inter-correlations were observed between isoprene OSs and NOSs (Table S4), suggesting their similar formation pathways via acid-catalyzed epoxide chemistry (Worton et al., 2013).

The average concentration of monoterpene OSs (α-pinene OSs, β-pinene OSs, limonene OSs and limonaketone OSs) was 0.6 ng/m$^3$, lower than those observed in mid-Atlantic (Meade et al., 2016) or the Pearl River Delta in southern China where more abundant emissions of BVOC precursors are expected (Wang et al., 2017d; He et al., 2014) (Table S3). The

contribution of monoterpene OSs was much lower than that of isoprene OSs or other OSs (Fig. 2, Table 1), as the mixing ratio of monoterpene (83 pptv) was lower than that of isoprene (297 pptv) during the campaign. Furthermore, the reactivity of monoterpenes with OH radical is lower than that of isoprene (Carlton et al., 2009; Paulot et al., 2009; Atkinson et al., 2006). Different from isoprene OSs, the four monoterpene OS species didn't show strong correlations with each other (Table S4), which may suggest their different oxidation mechanisms. While the contribution of monoterpene OSs was low, the monoterpene NOSs ($C_{10}H_{16}NO_7S^-$) were the second most abundant signals among the observed species (Table 1, Table S2), especially in the nighttime samples. The concentration of monoterpene NOSs ($C_{10}H_{16}NO_7S^-$) was much higher than those observed in mid-Atlantic or Belgium (Meade et al., 2016; Gómez-González et al., 2012), while lower than that observed in Pearl River Delta, South China (He et al., 2014). $C_{10}H_{16}NO_7S^-$ was also identified to be among the highest peaks in the mass spectra recorded by Orbitrap MS (Fig. 1 (b)), with a RI of 83% in the sample of 05/30N (Table S2). The monoterpene NOSs could be formed via nighttime $NO_3$-initiated oxidation under high-$NO_x$ conditions (Surratt et al., 2008; Iinuma et al., 2007; Gomez-Gonzalez et al., 2008). During the observation, both monoterpenes and $NO_x$ showed higher mixing ratios at night (Fig. S5 (a), (d)), favorable for the $NO_3$-initiated formation of NOSs.

**3.3 OS formation via acid-catalyzed aqueous-phase chemistry**

The time series of the total OS concentrations quantified by HPLC-MS are shown in Fig. 3, along with the meteorological conditions, $SO_2$, aerosol LWC, acidity, $PM_{2.5}$ and the major chemical components. Most OS species showed similar trends to the total OSs (Fig. S6), except for α-pinene OSs and β-pinene OSs, observed at very low concentrations. During the campaign, particles were generally acidic with a pH range of 2.0- 3.7, favorable for OS formation (Fig. 3). The aerosol acidity is indicated by aqueous phase [$H^+$] in this study. The OS concentrations generally followed a similar trend to that of sulfate aerosols (Fig. 3). The total OS concentrations showed strong correlations with sulfate (r=0.67) or aerosol acidity (r=0.67), suggesting the driving role of acidic sulfate aerosols in the OS formation (Table S4).

During the observation period, three pollution episodes (episodes I, II, III) were identified based on the $PM_{2.5}$ concentrations, which are marked by gray shadow in Fig. 3. The back trajectories, average concentrations of VOC precursors and oxidants during each episode are also shown in Table S5. The most significant increase trend of OSs was observed

during pollution episode III (nighttime of May 29 - the nighttime of May 30). During this episode, the accumulation of
secondary inorganic aerosols (SIAs), referring to sulfate, nitrate and ammonium in this study, was dominated by sulfate.
SIAs, especially sulfate and nitrate salts, represent the most important components driving the particle hygroscopicity (Wu et
al., 2018; Xue et al., 2014), thus the aerosol LWC increased with SIAs (Fig. 3). The increase of aerosol acidity was also
observed during this episode (Fig. 3). OSs increased to the highest level (129.2 ng/m$^3$) during the campaign under the
condition of high sulfate aerosols, high aerosol acidity and LWC (Fig. 3), suggesting the acid-catalyzed aqueous-phase
formation of OSs in the presence of acidic sulfate aerosols. The higher aerosol LWC encountered during these periods would
also favor the uptake of gas-phase reactants into particle phase, due to the decrease of viscosity and increase of diffusivity
within the particles (Shiraiwa et al., 2011). Moreover, the oxidant levels, indicated by $O_x$ ($NO_2$+$O_3$) in this study (Herndon et
al., 2008), were much higher than the other two episodes, which favored the formation of VOC oxidation products (e.g.
MVK+MACR) (Table S5). This is another reason for higher OSs concentration level during episode III. During pollution
episode II (nighttime of May 27 - the nighttime of May 28), the OS concentration level was lower than that during episode
III. It is noted that the increase of sulfate, aerosol LWC and acidity were also less than that during episode III, indicating less
aqueous-phase formation of OSs. During this episode, the increase of SIAs was attributed to both sulfate and nitrate, the two
with comparable contribution to the total SIAs. Different from episodes II and III, the SIAs accumulation was dominated by
nitrate during episode I (May 21- 23). OS and sulfate aerosols stayed at medium concentration level, lower than those during
the other two episodes. During the daytime of May 21, aerosol acidity increased due to the elevated relative contribution of
sulfate than that of nitrate, thus the OS concentration also increased. During the daytime of May 23, higher aerosol LWC
was observed due to the rapid increase of nitrate, however, the aerosol acidity was lower as a result of the less contribution
from sulfate. Thus, the increase of OS concentration was not very obvious. The OS formation may be limited by the aerosol
acidity, indicating the importance of acid-catalyzed chemistry. Stronger correlations between OSs and sulfate (r=0.67) or
aerosol acidity (r=0.67) compared with that between OSs and LWC (r=0.55) also suggest the importance of acid-catalyzed
chemistry for OSs formation. The back trajectories during episode I were different from those during episode II or III (Table
S5), which could be one reason for different conditions (e.g. SIA composition) during episode I. This episode ended with the
rain elimination event on the afternoon of May 23. The OSs were at low concentrations from May 24 to the daytime of May
27, when sulfate, $SO_2$, aerosol acidity and LWC were noticeably lower than the other periods, restraining the OS formation.
The three pollution episodes were characterized by different inorganic aerosol composition and aerosol properties (e.g.
acidity, LWC), resulting in different levels of OS formation. The concentrations and relative contribution of sulfate, aerosol
acidity and LWC are important factors influencing OS formation. The OS concentrations generally increased with the
increasing of sulfate, aerosol acidity and LWC (Fig. 3), suggesting more active OS formation via acid-catalyzed
aqueous-phase reactions in the presence of sulfate. These influencing factors were interrelated. Both sulfate and nitrate are
important hygroscopic components (Chan and Chan, 2005; Wu et al., 2018; Xue et al., 2014), favoring the water uptake of
aerosols and thus increasing LWC. The increasing of aerosol LWC with SIAs was observed (Fig. 3). A previous study also
suggested that at a given RH, aerosol LWC was nearly linearly related to the sum of nitrate and sulfate mass concentrations
(Guo et al., 2016). The variation of SIA composition and LWC would then influence the aerosol acidity (Liu et al., 2017;
Guo et al., 2016). In this study, higher aerosol acidity was observed with elevated contribution of sulfate among SIAs (Fig.
3). This is in accord with a previous study suggesting that particle pH was generally below 2 when aerosol anionic
composition was dominated by sulfate ($NO_3^-/2SO_4^{2-}$ mole ratio >1) (Guo et al., 2016).
To further elucidate the major factors influencing OS formation and their interrelations with SIA compositions, the
distribution of OS concentrations as a function of $SO_4^{2-}$/SIAs mass concentration ratios and other related factors are plotted
in Fig. 4. The aerosol LWC generally increased with the increasing of the SIA mass concentrations, while the aerosol acidity
was also influenced by the relative contribution of $SO_4^{2-}$ and $NO_3^-$ to SIAs. When the SIAs were dominated by $SO_4^{2-}$
($SO_4^{2-}$/SIAs> 0.5), the aerosol acidity increased obviously as a function of $SO_4^{2-}$/SIAs mass concentration ratios and the pH
values were generally below 2.8 (Fig. 4 (b, d)). The high aerosol acidity was favorable for OS formation and OS
concentration also increased as a function of sulfate mass concentration and fraction (Fig. 4 (a)). The pollution episode III
(Fig. 3) was the typical case for this condition. When the SIAs were dominated by nitrate ($SO_4^{2-}$/SIAs< 0.5), high LWC may
occur due to the high concentrations of hygroscopic SIAs, while the aerosol acidity was relatively lower due to the lower
sulfate fraction than that of nitrate (Fig. 4). The increase trend of OSs as a function of sulfate or $SO_4^{2-}$/SIAs mass
concentration ratios was not as obvious as the sulfate-dominant condition ($SO_4^{2-}$/SIAs> 0.5), as the OS formation may be
limited by lower aerosol acidity. The daytime of May 23 during pollution episode I (Fig. 3) was the typical case for this
atmospheric condition. Overall, the OS formation would obviously be promoted via acid-catalyzed aqueous-phase reactions,
when the SIAs accumulation was dominated by sulfate ($SO_4^{2-}$/SIAs> 0.5).

**3.4 Monoterpene NOS formation via the nighttime $NO_3$ oxidation**

A recent study suggested that nearly all the BVOCs could be oxidized overnight, dominated by reactions via $NO_3$
oxidation, at a $NO_x$/BVOCs ratio higher than 1.4 (Edwards et al., 2017). When we roughly estimated the BVOCs
concentration to be the sum of isoprene, MVK+MACR, and monoterpenes, the $NO_x$/BVOCs ratios were higher than 10 at
night (Fig. S5). This indicated the dominant nighttime BVOCs loss via $NO_3$-initiated oxidation in summer of Beijing. The
oxidation of BVOCs was found to be controlled by $NO_3$ oxidation rather than $O_3$ oxidation during the campaign, which
contributed to a total of 90% of BVOCs reactivity at night (Wang et al., 2018). Nighttime enhancement of monoterpene
NOSs was clearly observed under high-$NO_x$ conditions (Fig. 5). The nighttime concentrations of $C_{10}H_{16}NO_7S^-$ and
$C_9H_{14}NO_8S^-$ were respectively 1.3-31.4 (9.8 on average) and 0.9-19.7 (5.8 on average) times larger than daytime
concentrations. Higher mixing ratios of monoterpenes were observed at night (Fig. S5), when the high $NO_x$ concentrations
(Fig. 5) favored the formation of monoterpene NOSs via $NO_3$-initiated oxidation of monoterpenes. The elevated nighttime
concentrations of monoterpene NOSs was also observed in previous studies (Surratt et al., 2008; Iinuma et al., 2007;
Gomez-Gonzalez et al., 2008). High correlation between $N_2O_5$ and $NO_2$ or $NO_3$ radical production were observed (Wang et
al., 2018), so the $NO_2$ concentration was employed to investigate $NO_3$ oxidation during the campaign in this study. Higher
concentrations of monoterpene NOSs ($C_{10}H_{16}NO_7S^-$) were found with elevated $NO_2$ levels at night (Fig. 6), indicating the
plausibility of more NOS formation via $NO_3$-initiated oxidation. When $NO_2$ increased to higher than 20 ppb, the NOS
concentration did not further increase obviously with $NO_2$, which suggested that $NO_2$ was in excess and no longer the
limiting factor in NOS formation. The highest nighttime concentration of $C_{10}H_{16}NO_7S^-$ was recorded on May 27 during
episode II (Fig. 5). Besides the high $NO_2$ concentration (>20 ppb), the high monoterpene level was another primary reason
for the elevated concentration of monoterpene NOSs (Table S5).
The lower concentrations of monoterpene NOSs during the daytime could be attributed to the lower monoterpene, $NO_x$
and $NO_x$/BVOCs ratios than those at night (Fig. S5). What's more, monoterpene NOSs, also as organic nitrate (R-$ONO_2$)
compounds, may go through decomposition via photolysis or OH oxidation during the daytime (He et al., 2011;
Suarez-Bertoa et al., 2012). Organic nitrates have been estimated to have a short lifetime of several hours (Lee et al., 2016).
Elevation in concentrations of monoterpene NOSs were also observed with the increasing of $NO_2$ during daytime, but the
concentrations were much lower and the increase was less prominent than that during the nighttime (Fig. 6). The highest
daytime concentration of $C_{10}H_{16}NO_7S^-$ was recorded on May 23 (10.6 ng/m$^3$), followed by the daytime of May 31 (8.0
ng/m$^3$). The $NO_2$ concentrations were in the range of 20-25 ppb and 10-15 ppb during the daytime of May 23 and 31,
respectively. It is noted that the $J(O^1D)$ values during the daytime of May 23 and 31 were much lower than other daytime
periods (Fig. 5), indicating the possibility of less decomposition of monoterpene NOSs. Previous studies also reported that
the organic nitrate have much shorter lifetimes than the corresponding OSs, thus it is possible that organic nitrates derived
from monoterpene would undergo nucleophilic attack by sulfate and form monoterpene OSs or NOSs (He et al., 2014; Darer
et al., 2011; Hu et al., 2011). Monoterpene NOSs could also undergo hydrolysis and form monoterpene OSs (Darer et al.,
2011; Hu et al., 2011). These may be other potential pathways for the loss of monoterpene NOSs and production of
monoterpene OSs. These potential formation pathways of monoterpene OSs were different from the formation pathways via
acid-catalyzed aqueous-phase reactions. This could be another explanation for the different temporal variations of some
monoterpene OSs (Fig. S6) from other OSs.
**3.5 Formation pathways of isoprene OSs and NOSs**

430       Different from the day-night variation trend of monoterpene NOSs, isoprene NOSs ($C_5H_{11}NO_9S^-$) displayed similar

temporal variation to isoprene OSs and the total OSs (Fig. 7). Formation of the isoprene NOSs are supposed to have similar
limiting factors to those affecting isoprene OSs, rather than those limiting the nighttime $NO_3$-initiated formation of
monoterpene NOSs. The strong correlation between isoprene OSs and NOSs also indicated their similar formation pathways
or limiting factors in the formation (Table S4). The oxidation of isoprene could form isoprene epoxydiols (IEPOX),
hydroxymethyl-methyl-lactone (HMML) or methacrolein (MACR) and methacrylic acid epoxide (MAE) (Paulot et al., 2009;
Lin et al., 2013b; Worton et al., 2013; Nguyen et al., 2015). Both isoprene OSs and NOSs showed strong correlations with
isoprene oxidation products (MVK+MACR) (Table S4). The isoprene OSs could be formed through ring-opening epoxide
reactions of isoprene oxidation products, which was shown to be a kinetically feasible pathway (Minerath and Elrod, 2009;
Worton et al., 2013). Isoprene OSs were also proposed to form by reactive uptake and oxidation of MVK or MACR
(oxidation products of isoprene) initiated by the sulfate radicals (Nozière et al., 2010; Schindelka et al., 2013). Isoprene
NOSs generally increased with the increasing of isoprene oxidation products (MVK+MACR) and acidic sulfate aerosols
(Figs. 3 and 7, Table S4). It indicates isoprene NOSs form via acid-catalyzed reactions or reactive uptake of oxidation
products of isoprene by sulfate, rather than $NO_3$-initiated oxidation pathways. The highest concentrations of isoprene OSs
and NOSs were observed during the nighttime of May 30 during episode III (Fig. 7), with high sulfate, MVK+MACR,
aerosol acidity and LWC (Fig. 3, Table S5). In the formation of isoprene OSs or NOSs, epoxides first form carbocation
intermediates through acid-catalyzed hydrolysis reactions, and then sulfate ions serve as nucleophiles in the subsequent fast
step forming OSs or NOSs (Minerath and Elrod, 2009). The presence of high levels of sulfate may effectively facilitate the
ring-opening reaction of epoxide or reactive uptake of oxidation products and subsequent OSs or NOS formation (Surratt et
al., 2010). The proposed formation mechanisms of isoprene NOSs are needed to be further investigated and validated
through laboratory studies.

451        Although the isoprene NOS formation was not via the $NO_3$-initiated oxidation pathways, the $NO_3$ radical could be

involved in the formation pathways and influence the yield of isoprene NOSs. Considering the different atmospheric
conditions during the daytime and nighttime, we analyzed the variation of daytime and nighttime isoprene NOSs separately
(Fig. 8). Generally, higher concentrations of isoprene NOSs were found with elevated $NO_2$ or MVK+MACR concentration
levels. During daytime, the correlation of isoprene NOSs with $NO_2$ (r=0.74) was stronger than that with MVK+MACR
(r=0.69) (Fig. 8). When MVK+MACR was higher than 0.7 ppb, the NOS concentrations did not increase further with
MVK+MACR. It was likely that the biogenic VOCs precursors were in surplus under this condition and the formation of
isoprene NOSs may be limited by the lower daytime $NO_2$ concentration, sulfate aerosols or other factors. During daytime,
the MVK+MACR concentrations were generally higher and $NO_x$ was lower (Fig. S5), thus the $NO_2$ level may limit the
daytime formation of isoprene NOSs. During nighttime, a strong correlation between isoprene NOS and MVK+MACR
(r=0.94) was observed, while the increase trend of isoprene NOSs as a function of $NO_2$ (r=0.53) was not so obvious and their
correlation was lower (Fig. 8). During nighttime, the $NO_x$ concentrations were generally higher and MVK+MACR
concentrations were lower (Fig. S5), thus the concentrations of isoprene oxidation products (e.g. MVK+MACR) may be the
limiting factor for the nighttime formation of isoprene NOSs. The threshold (e.g. $NO_x$/isoprene ratio, $NO_x$/isoprene oxidation
products ratio) that makes the transition from $NO_x$-limited to isoprene-limited (or isoprene oxidation products) still need
further investigation through laboratory studies.

## 4    Conclusions

An intensive field campaign was conducted to investigate the characterization and formation of OSs and NOSs in
summer of Beijing, under the influence of abundant biogenic emissions and anthropogenic pollutants (e.g. $NO_x$, $SO_2$ and
$SO_4^{2-}$). The overall molecular characterization of S-containing organics (CHOS, CHONS) was made through ESI-Orbitrap
MS data. More than 90% of the CHOS formulas could be assigned as OSs and more than half of the CHONS formulas could
be assigned as NOSs, based on the molecular formulas. The number and intensity of OSs and NOSs increased significantly
during pollution episodes, which indicated they might play important roles for the SOA accumulation.
To further investigate the distribution and formation pathways of OSs and NOSs in complex ambient atmosphere, some
species were quantified using HPLC-MS, including ten OSs and three NOS species. The total concentrations of quantified
OSs and NOSs were 41.4 and 13.8 ng/m$^3$, respectively, accounting for 0.31% and 0.11% of organic matter. Glycolic acid
sulfate was the most abundant species (19.5 ng/m$^3$) among all the quantified OS species. The strong correlations between
GAS, LAS, HAS and isoprene OSs indicated their potential formation pathways via isoprene oxidation in the presence of
acidic sulfate aerosols. The concentration of isoprene OSs was 14.8 ng/m$^3$ and the isoprene OSs formed via $HO_2$ channel was
higher than that via $NO/NO_2$ channel. The contribution of monoterpene OSs was much smaller than other OSs, while the
monoterpene NOSs ($C_{10}H_{16}NO_7S^-$) were observed at high concentration (12.0 ng/m$^3$), especially in nighttime samples.
OS concentrations generally increased with the increase of acidic sulfate aerosols, aerosol acidity and LWC, indicating
the acid-catalyzed aqueous-phase formation of OSs in the presence of acidic sulfate aerosols as an effective formation
pathway. The sulfate concentration, SIA composition, aerosol acidity, and LWC are important factors influencing the OS
formation. When sulfate dominated the SIAs accumulation ($SO_4^{2-}$/SIAs> 0.5), the aerosol acidity would increase obviously
as a function of $SO_4^{2-}$/SIAs mass concentration ratios and the pH values were generally below 2.8. Thus, the OS formation
would be obviously promoted as the increasing of acidic sulfate aerosols, aerosol acidity and LWC. When the SIAs
accumulation were dominated by nitrate ($SO_4^{2-}$/SIAs< 0.5), high aerosol LWC may occur, while the OS formation via
acid-catalyzed reactions may be limited by relatively lower aerosol acidity.

490        The $NO_3$-initiated oxidation dominated the nighttime BVOCs loss in summertime Beijing, with the $NO_x$/BVOCs ratios

higher than 10 at night. Significant nighttime enhancement of monoterpene NOSs was observed, indicating the formation via
$NO_3$-initiated oxidation of monoterpene under high-$NO_x$ conditions. Higher concentrations of monoterpene NOSs were
found with elevated $NO_2$ levels at night and $NO_2$ ceased to be a limiting factor for NOS formation when higher than 20 ppb.
The lower daytime concentrations of monoterpene NOSs could be attributed to the lower production and the decomposition
during daytime. Different from the monoterpene NOS formation via $NO_3$-initiated oxidation, isoprene NOSs and OSs are
supposed to form via acid-catalyzed chemistry or reactive uptake of the oxidation products of isoprene, which is needed to
be further investigated through laboratory studies. The daytime $NO_2$ concentration could be a limiting factor for isoprene
NOS formation, while the nighttime formation was limited by isoprene or its oxidation products. The proposed formation
mechanisms of isoprene NOSs as well as the limiting factors still need further investigation in laboratory studies.

500        This study highlights the formation of OSs and NOSs via the interaction between biogenic VOC precursors and

anthropogenic pollutants ($NO_x$, $SO_2$ and $SO_4^{2-}$) in summer of Beijing. Our study reveals the accumulation of OSs with the
increase of acidic sulfate aerosols and the nighttime enhancement of monoterpene NOSs under high-$NO_x$ conditions. The
acidic sulfate aerosols and high nighttime $NO_x$ or $N_2O_5$ concentrations were observed in Beijing in our observation and also
other studies (Liu et al., 2017; Wang et al., 2017b; Wang et al., 2017a), which provide favorable conditions for the formation
of OSs and NOSs. The results imply the importance of reducing anthropogenic emissions, especially $NO_x$ and $SO_2$, to reduce
the biogenic SOA burden in Beijing, and also in areas with abundant biogenic emissions and anthropogenic pollutants.
Moreover, the OSs or NOSs could be treated as key SOA species when exploring the biogenic-anthropogenic interactions as
well as organic-inorganic reactions.

*Data availability*. The dataset is available upon request by contacting Min Hu (minhu@pku.edu.cn).

**The Supplement related to this article is available online**

*Competing interests*. The authors declare that they have no conflict of interest.

*Acknowledgements*. This work was supported by National Natural Science Foundation of China (91544214, 41421064,
51636003); National research program for key issues in air pollution control (DQGG0103); National Key Research and
Development Program of China (2016YFC0202000: Task 3); bilateral Sweden-China framework program on
'Photochemical smog in China: formation, transformation, impact and abatement strategies' (639-2013-6917).

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

**Figures**

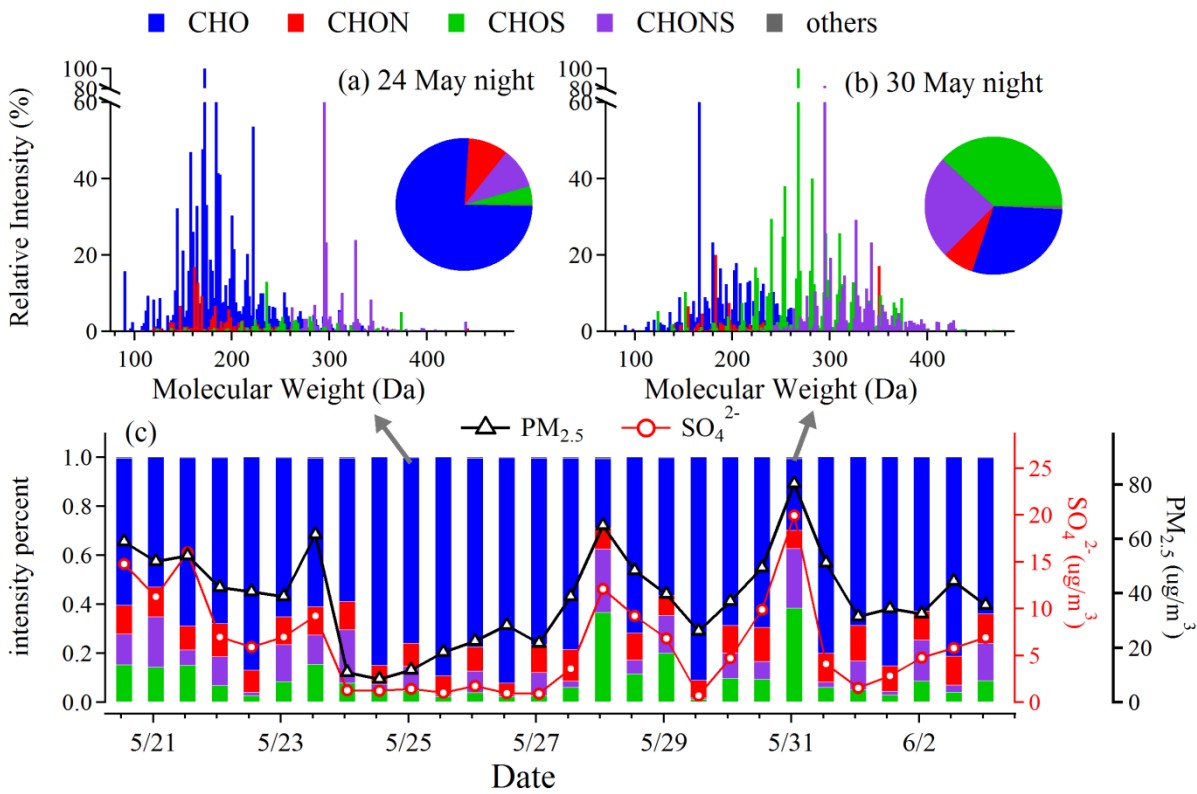

Figure 1 The intensity distribution of different compound categories (CHO, CHON, CHOS and CHONS) (a) on a clean day and (b) on a polluted day. (c) Temporal variation of $PM_{2.5}$, $SO_4^{2-}$ and intensity percentages of different compound categories. The highly water-soluble OS species (e.g. isoprene OSs) with lower MW are absent in these figures and details are described in section 3.1.

887

888

889

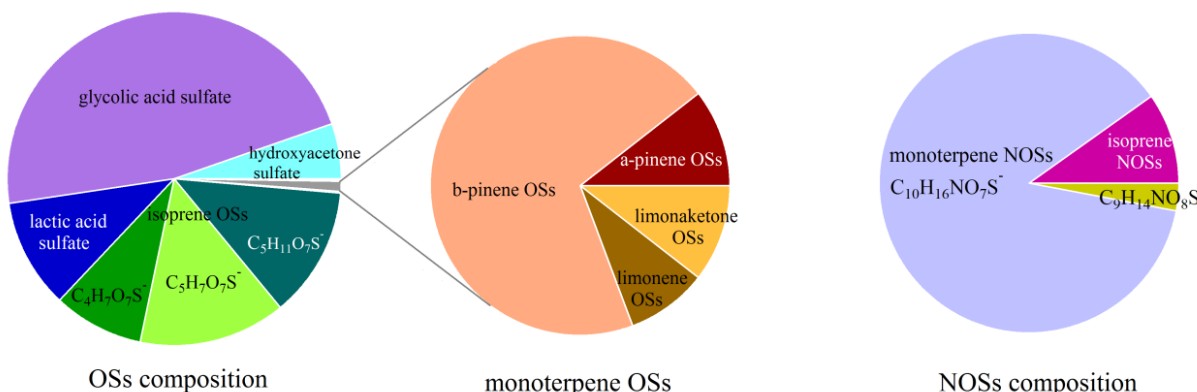

OSs composition    monoterpene OSs    NOSs composition

Figure 2 The relative contribution of different OS and NOS species. Only the selected species (semi-)quantified by

HPLC-MS are included in this figure.

893

894

895

896

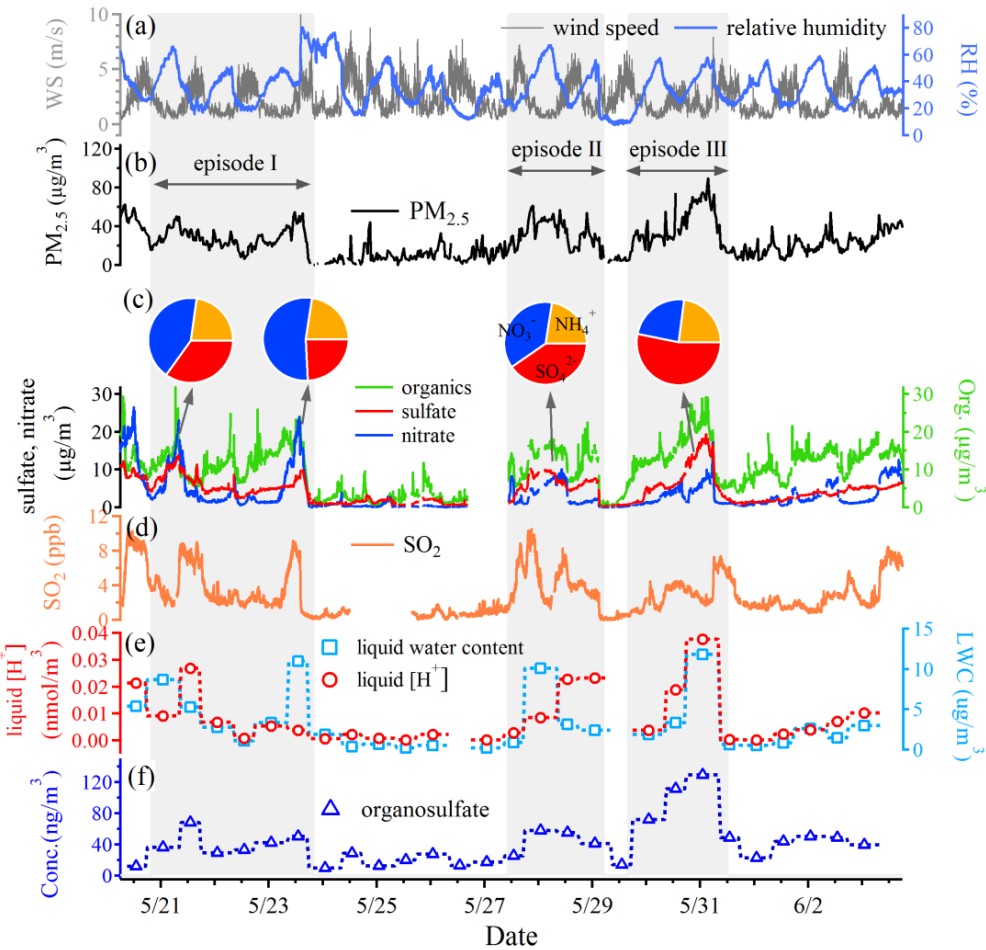

Figure 3 Time series of (a) wind speed (WS) and relative humidity (RH), (b) PM$_{2.5}$, (c) mass concentrations of organics, sulfate, nitrate and composition of secondary inorganic aerosols during pollution episodes (d) SO$_2$, (e) liquid water content (LWC) and aqueous phase [H$^+$], and (f) the total concentrations of OSs quantified by HPLC-MS. The pollution episodes were marked by gray shadow.





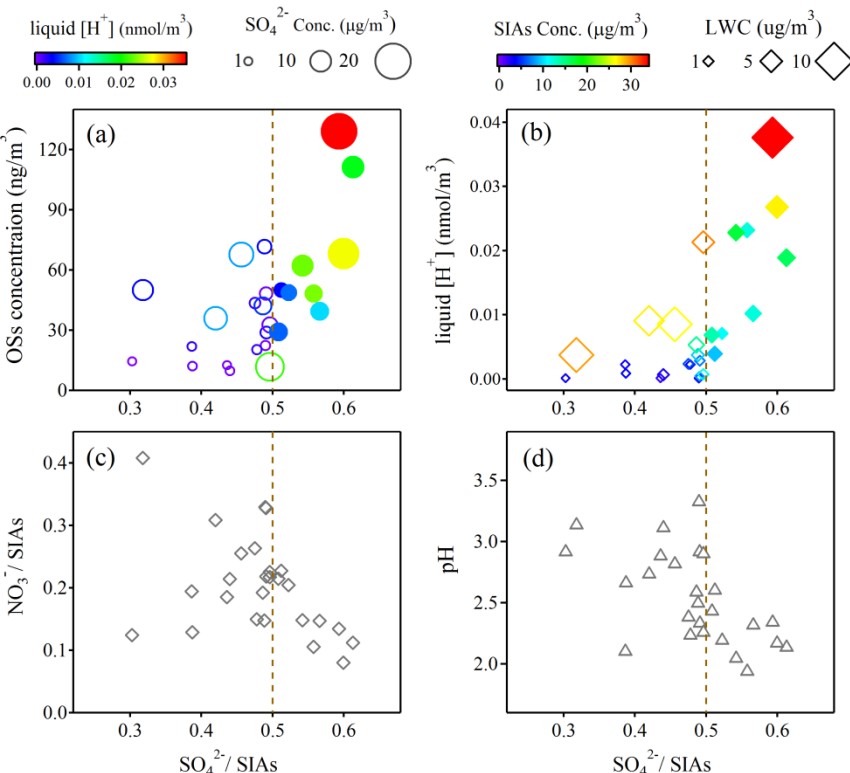


Figure 4 (a) The OS concentrations as a function of the $SO_4^{2-}$/ SIAs mass ratios. The circles are colored according to the
liquid [H$^+$] concentration and the sizes of the circles are scaled to the $SO_4^{2-}$ mass concentration. (b) The liquid [H$^+$] as a
function of the $SO_4^{2-}$/ SIAs mass ratios. The markers are colored according to the SIAs mass concentrations and the sizes of
the markers are scaled to the liquid water content (LWC). (c) The $NO_3^-$/ SIAs mass ratios as a function of the $SO_4^{2-}$/ SIAs
mass ratios. (d) The aerosol pH as a function of the $SO_4^{2-}$/ SIAs mass ratios. The solid markers represent those among the
range $SO_4^{2-}$/ SIAs > 0.5 and hollow markers represent those among the range $SO_4^{2-}$/ SIAs < 0.5 in figure (a) and (b). When
sulfate dominated the accumulation of secondary inorganic aerosols ($SO_4^{2-}$/SIAs> 0.5), both aerosol LWC and acidity (pH<
2.8) increased and OS formation was obviously promoted. In comparison, the acid-catalyzed OS formation was limited by
lower aerosol acidity under nitrate-dominant conditions.




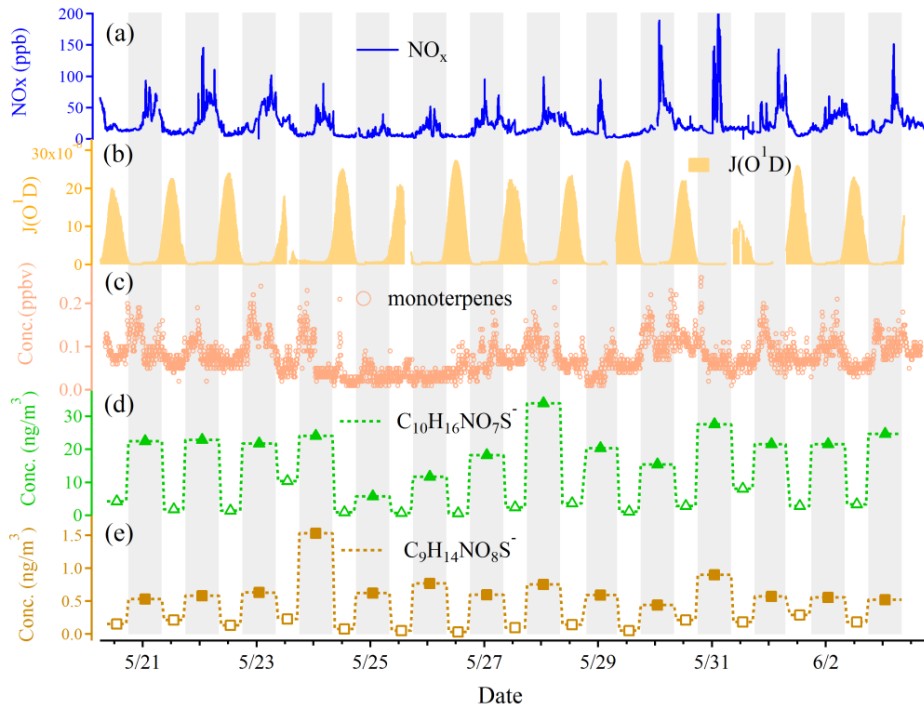


Figure 5 Time series of (a) $NO_x$, (b) $J(O^1D)$, (c) monoterpene, (d) monoterpene NOSs ($C_{10}H_{16}NO_7S^-$) and (e) limonaketone

NOSs ($C_9H_{14}NO_8S^-$). The gray background denotes the nighttime and white background denotes the daytime.




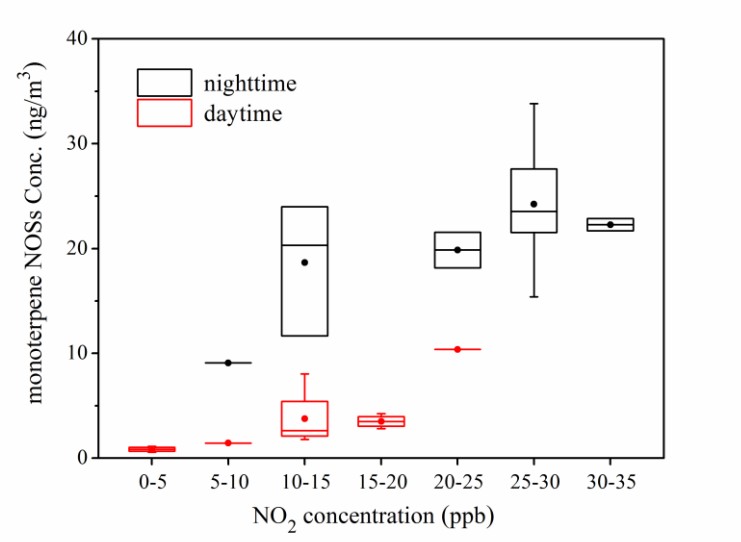


Figure 6 The concentrations of monoterpene NOSs ($C_{10}H_{16}NO_7S^-$) as a function of $NO_2$ concentration bins (ppb) during
daytime and nighttime. The closed circles represent the mean values and whiskers represent 25 and 75 percentiles.


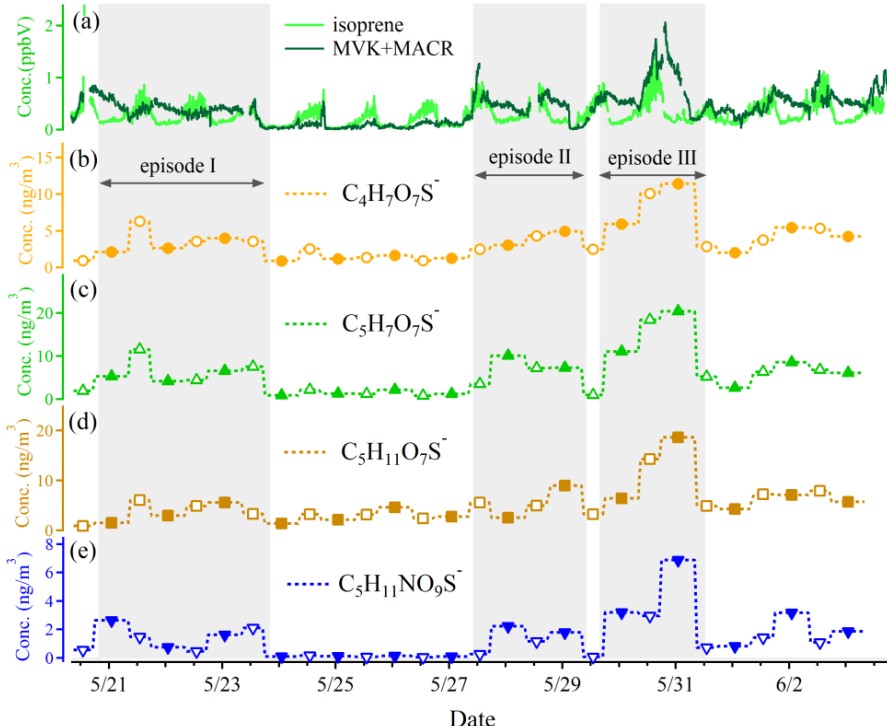


Figure 7 Time series of (a) isoprene and MVK+MACR, isoprene OSs (b) $C_4H_7O_7S^-$, (c) $C_5H_7O_7S^-$, (d) $C_5H_{11}O_7S^-$ and (e)

NOSs ($C_5H_{11}NO_9S^-$). The pollution episodes were marked by gray shadow. MVK and MACR are the abbreviations of

methyl vinyl ketone and methacrolein, respectively.


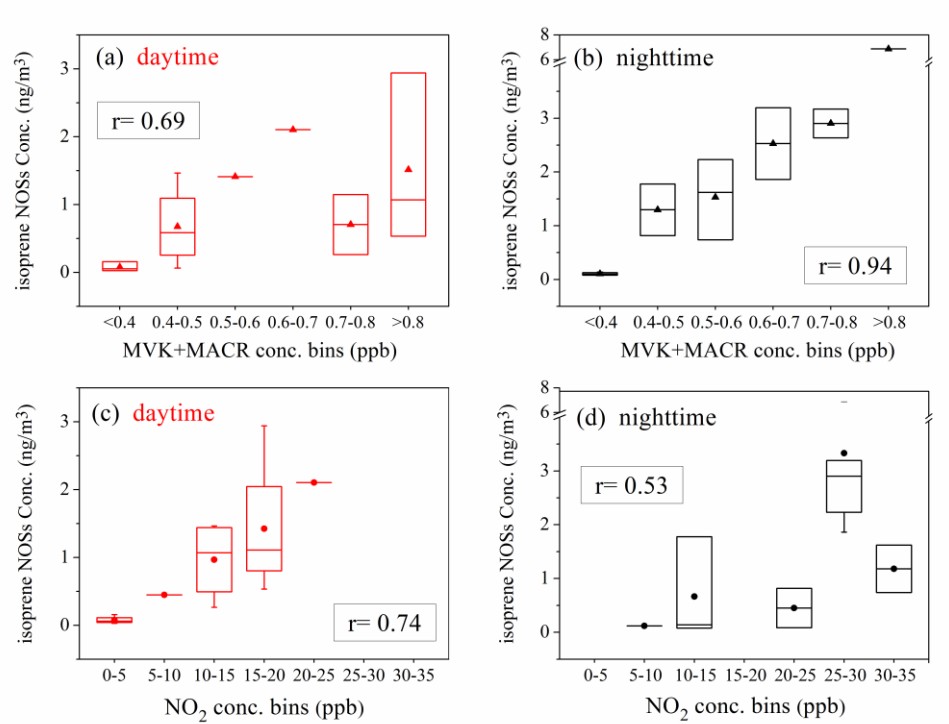


Figure 8 The isoprene NOSs ($C_5H_{11}NO_9S^-$) concentrations as a function of $NO_2$ or MVK+MACR concentration bins (ppb) and the correlations between isoprene NOSs ($C_5H_{11}NO_9S^-$) and $NO_2$ or MVK+MACR. The closed markers in the box represent the mean values and whiskers represent 25 and 75 percentiles in each concentration bin. The r value in each panel represents the correlation coefficient between isoprene NOSs and $NO_2$ or MVK+MACR concentrations.

940

941

**Tables**

Table 1 Organosulfates and nitrooxy-organosulfates quantified by HPLC-MS

| common name | formula | [M-H]⁻ | retention time (min) | standard | structure | concentration (ng/m³) | |
|---|---|---|---|---|---|---|---|
| | | | | | | range | average (n=28) |
| Hydroxyacetone sulfate (HAS) | $C_3H_5O_5S^-$ | 152.99 | 1.7, 2.5 | Glycolic acid sulfate |  (Hettiyadura et al., 2015) | 0.5-7.5 | 2.2 |
| Glycolic acid sulfate (GAS) | $C_2H_3O_6S^-$ | 154.97 | 1.6, 2.3 | Glycolic acid sulfate |  (Olson et al., 2011) | 3.9-58.2 | 19.5 |
| Lactic acid sulfate (LAS) | $C_3H_5O_6S^-$ | 168.98 | 1.6, 2.6 | Lactic acid sulfate |  (Olson et al., 2011) | 0.7-11.9 | 4.4 |
| Isoprene OSs | $C_4H_7O_7S^-$ | 198.99 | 1.5, 2.9 | Lactic acid sulfate |  (Lin et al., 2013b; Surratt et al., 2007; Hettiyadura et al., 2015) | 0.9-11.4 | 3.6 |
| | $C_5H_7O_7S^-$ | 210.99 | 1.8, 2.9 | Lactic acid sulfate |  (Surratt et al., 2008; Hettiyadura et al., 2015) | 0.8-20.4 | 5.9 |
| | $C_5H_{11}O_7S^-$ | 215.02 | 1.6, 2.0 | Lactic acid sulfate |  (He et al., 2014; Surratt et al., 2008) | 0.9-18.7 | 5.3 |

| Name | Formula | m/z | RT | Proposed structure name | Structure | Range | Mean |
|---|---|---|---|---|---|---|---|
| Isoprene NOS | $C_5H_{10}NO_9S^-$ | 260.01 | 4.9 | Lactic acid sulfate | (Surratt et al., 2007) | 0.03-6.9 | 1.4 |
| α-pinene OS | $C_{10}H_{17}O_5S^-$ | 249.08 | 22.7 | α-pinene OS | (Wang et al., 2017d; Surratt et al., 2008) | 0.01-0.5 | 0.06 |
| β-pinene OS | $C_{10}H_{17}O_5S^-$ | 249.08 | 22.4, 23.4 | β-pinene OS | (Wang et al., 2017d; Surratt et al., 2008) | 0.07-0.8 | 0.4 |
| Limonene OS | $C_{10}H_{17}O_5S^-$ | 249.08 | 21.8, 23.8 | Limonene OS | (Wang et al., 2017d) | 0.01-0.1 | 0.05 |
| Limonaketone OS | $C_9H_{15}O_6S^-$ | 251.06 | 14.0 | Limonaketone OS | (Wang et al., 2017d) | 0.00-0.2 | 0.06 |
| Monoterpene NOSs | $C_{10}H_{16}NO_7S^-$ | 294.06 | 24.8, 26.6, 27.1 | α-pinene OSs | (Surratt et al., 2008; He et al., 2014) | 0.6-33.8 | 12.0 |
|  | $C_9H_{14}NO_8S^-$ | 296.04 | 21.1 | Limonaketone OS | (Surratt et al., 2008) | 0.03-1.5 | 0.4 |