# Peer review of "between Biogenic Emissions and Anthropogenic Pollutants in"

_Atmospheric Chemistry and Physics, 2018_

## Referee Comment (RC1) · Anonymous Referee #1 · 13 Apr 2018

This paper describes an observation of organosulfates (OSs) as well as nitroxy-organosulfates (NOSs) in Beijing over a summer period. A number of OSs and NOSs are detected by direct infusion electrospray mass spectrometry. In addition, several OSs are quantified by HPLC-ESI methods using authentic standards or surrogates. The measurements are very done and cover a meaningfully long period of time making it possible to qualitatively correlate the presence of OSs or NOSs with other environmental conditions, such as precursor concentrations, liquid water content, particle acidity etc. The paper should be published after some minor improvements described

below.

In my opinion, the introduction section does not explain the motivation of the study sufficiently well. The introduction sections states that OSs are important, were observed many times, can form by mechanisms that are affected by environmental conditions, and can now be in some cases quantified with authentic standards. However, it does not explain what this study is trying to accomplish and how it builds on all the previous field and lab work on OSs and NOSs. There is a clue about what authors want to accomplish in the sentence on lines 89-90, but this is not enough. Ideally, a testable hypothesis or a clear set of goals should be posed in the introduction.

Another issue I have with the introduction is that it motivates the study by saying that we cannot predict the amount of SOA correctly. The implication is that formation of rather involatile OSs and NOSs should help resolve this discrepancy. However, the mass concentrations of OSs reported here are rather small compared to the mass concentration of organic matter in particles. Since OSs and NOSs are minor species, it is better to motivate the study by our quest to understand the acid-catalyzed chemistry in particles, of which formation of OSs is an example, and the night time chemistry of NO3 in urban environments, which appears to be partly responsible for the observed NOSs.

I found the discussion in sections 3.3 and 3.5 too qualitative. I realize that the data set mat be not long enough to make more definitive conclusions. Tied to this is Figure 4, the data in which are too scattered to make any reliable conclusions. I am not sure what the authors can do about it under the circumstances. Is Figure 4 the best way to look for correlation in the data? Have the authors tried correlating the amount of observed OSs to, for example, a product of sulfate, organic mass, and hydronium ion concentrations that might be expected to describe an acid-catalyzed reaction? I would encourage them to come up with more convincing ways of presenting the data then currently afforded by Figure 4.

Here are minor editorial comments.

L57-58: this is an unclear sentence, please revise

L104: monoterpene -> monoterpenes

L124: mass resolution -> mass resolving power

L128:please be more explicit about the allowed elements and constraints placed on the formulas

L145: OSs and NOS species -> OS and NOS species

L149: Olson et al. (2011) (Olson et al., 2011) -> Olson et al. (2011). Similar corrections may be needed in other places in the manuscript, for example, on line 153

L171: please clarify what are "compounds excluded from the above major compound categories". Would it be CHS? Or peaks that could not be assigned within the imposed constraints? It would alos be useful to know what fraction of peaks was assigned.

L202: OSs molecules -> OS molecules

L202: monoterpene -> monoterpenes

L203: molecules -> ions

L205: relatively higher relative intensity -> higher relative intensity

L206: less -> fewer

L221: does the "total concentrations of quantified OSs" refer the average over all the samples?

L227: Where is "Centreville" located? (Unlike the other locations mentioned, it is not a country or state)

L268: monoterpene -> monoterpenes

L270: were -> was

L289 and elsewhere in this section: Secondary inorganic aerosols (SIAs) are defined as sulfate, nitrate and ammonium on line 289. However, the ratio of SO4(2-) to SIAs above or below 0.5 is then used to separate conditions into sulfate and nitrate dominated regimes. The exact definition of this ratio is not clear. To be more precise the authors should define a molar ratio SO4(2-)/[SO4(2-)+NO3(-)] and use it in their discussion instead of the vaguely defined ratio they are currently using.

L304-305: please fix grammar in this sentence

L345 and 352: the "nighttime formation" and "lower production" are not plausible explanations. The diurnal profile of monoterpenes may be peaking at night because they are removed more slowly at night and/or boundary layer height is changing. It should say "nighttime peak in concentrations" instead of "nighttime formation".

Table 1: some of the OS structures are draw as ionized (deprotonated) and some are not. Since the formulas are given for the ionized forms, it would be good to draw the structures for the ionized forms as well for consistency
* * *

---

## Referee Comment (RC2) · Anonymous Referee #2 · 25 Apr 2018

This study investigates the composition, abundance, and formation pathways of organosulfates during the day and night in the summer of Beijing under the influence of biogenic and anthropogenic emissions. Under various pollution episodes characterized by different aerosol composition and levels of gas-phase pollutants, the authors show that the formation of organosulfates can be largely influenced by the concentrations and relative contribution of sulfate, aerosol acidity and liquid water content. This work provides new field observation data to better understand the abundance and formation of organosulfates in the atmosphere. I support the publication of this paper after

addressing the following questions.

Major comments:

My major comment is the calculation and determination of the aerosol acidity and aerosol phase water content.

Line 161, "Aqueous phase [H+] and LWC were then calculated with the ISORROPIA-II thermodynamic model. ISORROPIA-II was operated in forward mode, assuming the particles are "metastable" (Hennigan et al., 2015; Weber et al., 2016; Guo et al., 2015). The input parameters included: ambient RH, temperature, particle phase inorganic species (SO42-, NO3-, Cl-, NH4+, K+, Na+, Ca2+, Ma2+) and gaseous NH3."

First, the authors should provide justifications why ambient aerosols can be assumed to be "metastable" in this work. It is important to discuss the physical state of the ambient aerosols (e.g. solid, liquid or solid/liquid) during the field campaign. This information is important for analyzing and interpreting the data given aqueous aerosol-phase reactions have been proposed as one of the major formation pathways for the organosulfates in this work. Some major conclusions are drawn based on this assumption. A detailed explanation is needed

Second, given the organic compounds contribute significantly to the total aerosol mass in this work, do the organic compounds being considered when the aerosol acidity and aerosol water content are calculated using the aerosol thermodynamic model. If not, the authors should discuss how the organic compounds would affect the predictions of the aerosol acidity and aerosol water content. Would the findings or conclusions drawn from the data based on inorganic species only affect by the presence of organic compounds? The authors should discuss how they determine the ambient RH and temperature used for their model simulations.

Minor comments:

Line 147 " OSs and NOSs were quantified using authentic standards or surrogates
with similar molecular structures (Table 1). Lactic acid sulfate (LAS) and glycolic acid sulfate (GAS) were prepared according to Olson et al. (2011) (Olson et al., 2011). Four monoterpene derived OSs were synthesized according to Wang et al. (2017) (Wang et al., 2017d)." What is the purity of these standards used in this work? What is the recovery or extraction efficiency of these standards?

Line 172, "The OC content in each sample for Orbitrap MS analysis was kept roughly constant to minimize variation arising from matrix ion suppression." Please elaborate how to achieve this goal.

Line 177, "What's more, the S-containing compounds contributed more to the higher MW formulas than CHO (O1-O10) or CHON (O1-O11) compounds (Fig. 1), due to the existence of more O (CHOS: O1-O12, CHONS: O1-O14) atoms and heteroatoms (S, N) in the molecules. They may play more important roles in the increase of SOA mass concentrations during pollution episodes." Given the concentration of quantified organosulfates is small, the formation of organosulfates is not likely explained the increase of SOA mass. Please elaborate this point.

Line 233, "carboxylic acids mainly form via aqueous phase oxidation in cloud or particle water, including both biogenic and anthropogenic sources (Charbouillot et al., 2012;Chebbi and Carlier, 1996). The relatively higher level of hydroxycarboxylic acid sulfate could be attributed to the favorable interaction between sulfate aerosols and carboxylic acids or other precursors in summertime Beijing, while the precursors and mechanisms remain unclear." The authors should provide more information or field data to support this argument. For example, what are the concentrations of these carboxylic acids if these acids have been measured in this field campaign?

Line 260 "unclear. The concentration of isoprene NOSs (C5H10NO9S-) was lower than that of individual isoprene OSs. Strong inter-correlations were observed between isoprene OSs and NOSs (Table S2), suggesting their similar formation pathways." Please elaborate what are the formation pathways.
For the 3.3 OS formation via acid-catalyzed aqueous phase chemistry, I understand the authors focus on understanding how the quantified organosulfates form under different pollution episodes. Do the authors observe the same results for the "unquantified" organosulfates (or other detected organosulfates) as well?

Line 294 "Moreover, the higher aerosol LWC encountered during these periods would favor the uptake of gas-phase reactants into particle phase, due to the decrease of viscosity and increase of diffusivity within the particles (Shiraiwa et al., 2011)." These descriptions would be too qualitative. What would be the viscosity of the ambient aerosols expected in this work? How would the aerosol viscosity change with the aerosol composition and environmental conditions (e.g. ambient temperature and RH) in this work?

Line 321, in the section, the authors elucidate the major factors influencing the formation of quantified organosulfates and their interrelations with SIA compositions. Do the authors also observe the same results for the unquantified organosulfates (or other detected organosulfates)?

**ACPD**

---

## Referee Comment (RC3) · Anonymous Referee #3 · 4 May 2018

This manuscript describes measurements of organosulfates and related compounds in Beijing during May-June 2016. Emphasis is placed on understanding the factors that influence organosulfate and nitrooxyorganosulfate formation, particularly during three pollution episodes. The manuscript concludes that sulfate, liquid water content, and acidity are important factors in their formation.

The measurements appear to be carefully conducted and well-described. However, there are some shortcomings in the presentation of the data that should be addressed prior to publication.

[Figure]

Major comments:

1) The discussion of the trends in organosulfates and co-located measurements in section 3.3 is limited to qualitative descriptions. Correlation analysis (like that conducted between observed organosulfates, Table S2) should be extended to include co-located measurements of sulfate, nitrate, ammonium, liquid water content, aerosol acidity [H+], VOC precursors, oxidants, etc. to provide quantitative support for the associations (or lack thereof) that are discussed in this section.

2) Correlation analysis should also be conducted and presented to support the discussion in section 3.5.

3) The phrase "representative organosulfates" is used in several instances (line 22, 218, 411) although the authors do not indicate what these species represent. Rather than using this vague language, the authors should more explicit in describing why the selected compounds were quantified and semi-quantified.

4) The discussion in the paragraph beginning at line 209 implies that the only difference between the three air pollution episodes was their inorganic ion content (which affected aerosol acidity and liquid water content). Do back trajectories, VOC concentrations, and other co-located measurements support this? If not, how could variations in other atmospheric conditions explain the organosulfate observations?

5) The overall concentration of organosulfates observed in Beijing seems to be very low (~150 ng/m3). In encourage the authors to discuss this observation and include it in their comparison to prior studies.

6) A table comparing key species and total organosulfate concentrations across this and prior studies would be a useful addition to the supplement to support the comparison of data.

7) In two places (line 117 and 173) the authors indicate that the organic carbon concentration was held constant across samples analyzed by Orbitrap, in order to decrease

ion suppression. The authors should provide a reference to support this statement and/or evidence to support that ion suppression did not occur.

8) The SPE method described at line 118 indicates that select compounds are removed by the SPE process. However, there is no mention until line 207-215 what effect the SPE has on organosulfates. I suggest making a note at line 120 indicating that further discussion of the impact of this clean-up procedure on organosulfates is discussed in section 3.1.

9) Line 150 – please explain how semi-quantification is achieved for a surrogate standard that appears to be comprised of a mixture of compounds (e.g., "alpha-pinene OS").

10) The discussion at lines 198-202 implicates long-chain alkanes and diesel/biodiesel emissions as the source of several organosulfates. Can the authors please comment on the (un)certainty of these assignments and the possibility that they may derive from monoterpenes (given the similarities in the molecular formulas to the monoterpene-derived organosulfates mentioned later in the same paragraph)?

11) Line 225-227, please include the city, state, and country for each of the measurement sides discussed. Centreville and summertime Alabama are presented as though they are different locations, when they are one in the same.

12) I encourage the authors to consider their use of significant figures in reporting their data. Many organosulfate concentrations are listed to four significant figures, while their contributions to organic carbon have only one. The former seems to be too many (considering measurement uncertainties and use of surrogate standards) and the latter seems to be not enough.

13) In Table S2, please label which compounds are "isoprene OS" that are mentioned at line 238.

14) In several places, the wording should be adjusted so as to better reflect that many

species were semi-quantified and absolute concentrations remain unknown. At line 244: "The concentration of quantified isoprene OS..." At line 271 "...were the second most abundant signals among the observed species..."

15) In Figure 1, please write out the dates "24 May night" and "30 May night" rather than "0524N" and "0530N".

16) In the figure 2 caption, please point the reader to the specific section where the information about the missing water-soluble OS can be found (3.1)

17) In the Figure 2 caption, please explain that these plots only include the select species quantified or semi-quantified by LCMS.

18) There is a lot going on in Figure 4. Can this be simplified? Or perhaps broken into multiple graphs that do not have so much overlap? Also, because there is so much going on, adding the key findings / takeaway messages from the graphs to the caption would help the reader.

19) Figure S1 – delete 2016 from the date on the x-axis (as this takes up unnecessary room). It would be helpful to designate what is daytime and nighttime in this figures as is done in many of the other figures.

20) Table S1 would benefit from organization by m/z so that the table can be easily navigated by other researchers who are likely to look up the data in this way.

Technical/editorial comments

21) Line 20: "the majority"

22) Line 22: "mass spectrometry was employed"

23) Define LWC at line 27

24) Define SIA species at line 28

25) "0.02%" of OA at line 46 seems too small, is this reasonable?

26) Line 109: "flow rates were"

27) Line 164: Mg2+ (rather than Ma2+)

28) Line 172: do not need to say "percent" in either instance, since it is earlier in the sentence.

29) Line 283: "favorable for OS formation"

30) Hettiyadura et al. (2015) propose a mechanism for the formation of the isoprene organosulfate with m/z 211 that is consistent with the hypothesis presented by Surratt et al. (2008).

31) Line 343: "times larger than daytime"

32) Line 349: "levels at night..."

33) Line 351: "was in excess and no longer the limiting factor in NOS formation.

34) Line 380: "NOSs form via"

35) Line 382: "formation of isoprene OSs or NOSs, epoxides first form..."

36) Line 393: "increase further with MVK+MACR."

37) Line 419: "OS concentrations"

38) Line 430: "NO2 levels at night..."

---

## Referee Comment (RC4) · Anonymous Referee #4 · 4 May 2018

**Summary**

This ACPD article characterizes the amount of organosulfates (OSs) and nitrooxy organosulfates (NOS) through ESI-Orbitrap and HPLC-MS during a field campaign near Beijing, China. By using standards or surrogates, the manuscript breaks down organosulfates into isoprene-derived OSs and monoterpene derived OSs, and shows isoprene-derived OSs dominated the total OSs.

By analyzing inorganic aerosol composition, acidity, and liquid water content, the author concludes that due to acid catalyzed chemistry, the production of isoprene-derived OSs was strongly correlated with the acidity of the particles, which was governed by sulfate percentage in secondary inorganic aerosols.

The monoterpene-derived NOSs and isoprene-derived NOSs were measured mainly by the HPLC-MS and their concentrations were used to correlate with the ambient NOx concentration. Monoterpene NOSs were greatly enhanced during night time due to high NOx concentration.

As written in the manuscript, through measurements of OSs and NOSs, this study describes the interaction between biogenic emission and anthropogenic pollutants. The data of this kind are valuable and fits into the scopes of ACP. Overall, the manuscript is sound and after addressing the following issues, it is suitable to be published on ACP.

**Comments**

Line 145: This part was not very clear. How did the author obtain the monoterpene NOSs for quantification? Was it synthesized or commercially available? Please illustrate in detail.

Line 239: The manuscript describes the strong correlation between GAS, LAS, and HAS with isoprene OSs. Then the author concludes that "isoprene or its oxidized products as potential precursors of GAS, LAS and HAS." The logic here is flawed. Isoprene OSs are correlated with sulfate concentration. Therefore, it is very much likely that GAS, LAS, and HAS are just correlated with sulfate concentration. The correlation of GAS with isoprene OSs does not mean isoprene oxidation products may be precursors of GAS, LAS, and HAS. I suggest changing this part to: "They also showed strong correlations with isoprene OSs (Table S2), suggesting GAS, LAS, and HAS can be potential tracers for organosulfates."

Line 250: The author raised a very interesting point here. The Southeast U.S. has an isoprene OSs concentration of 165 ng cm$^{-3}$ (Rattanavaraha et al., 2016). The isoprene concentration in Beijing is only 5 times lower than Southeast U.S, but the isoprene-derived OS is 10 times lower. The average RH in Beijing is also lower than Southeast U.S. Maybe the author should provide this evidence to further support the statement that organic coatings and their phase can play an important role.

There are a few important references that I would suggest adding to the manuscript:
Line 69: I suggest adding Shrestha et al., 2014; Zhang et al., 2015 to provide more evidence for how RH and LWC affect aerosol viscosity.
Line 66: I suggest adding Riedel et al., 2015 to show acid-catalyzed reactive uptake reactions in forming isoprene-derived OA.
Line 250: I suggest adding Riva et al., 2016 to show the effects of pre-existing organic loading on isoprene-derived SOA formation. Riva et al. was the first to show the effect of OA on isoprene SOA formation, and Zhang et al. 2018 was the first to quantify such effects.

Line 164: Oxford comma is suggested here.

Line 371-372: Are there any evidence to show that isoprene NOSs are formed via similar pathways (or multiphase reactions) as isoprene OSs? To my knowledge there is limited experimental evidence to show the isoprene NOSs are formed through multiphase reactions. The author should provide more evidence to support the argument.

**References**

Rattanavaraha, W., Chu, K., Budisulistiorini, S. H., Riva, M., Lin, Y. H., Edgerton, E. S., Baumann, K., Shaw, S. L., Guo, H., King, L., Weber, R. J., Neff, M. E., Stone, E. A., Offenberg, J. H., Zhang, Z., Gold, A., and Surratt, J. D.: Assessing the impact of anthropogenic pollution on isoprene-derived secondary organic aerosol formation in PM2.5 collected from the Birmingham, Alabama, ground site during the 2013 Southern

Oxidant and Aerosol Study, Atmos. Chem. Phys., 16, 4897-4914, 10.5194/acp-16-4897-2016, 2016.

Riedel, T. P., Lin, Y.-H., Budisulistiorini, S. H., Gaston, C. J., Thornton, J. A., Zhang, Z., Vizuete, W., Gold, A., and Surratt, J. D.: Heterogeneous Reactions of Isoprene-Derived Epoxides: Reaction Probabilities and Molar Secondary Organic Aerosol Yield Estimates, Environ. Sci. Technol. Lett., 2, 38-42, 10.1021/ez500406f, 2015.

Riva, M., Bell, D. M., Hansen, A.-M. K., Drozd, G. T., Zhang, Z., Gold, A., Imre, D., Surratt, J. D., Glasius, M., and Zelenyuk, A.: Effect of Organic Coatings, Humidity and Aerosol Acidity on Multiphase Chemistry of Isoprene Epoxydiols, Environ. Sci. Technol., 50, 5580-5588, 10.1021/acs.est.5b06050, 2016.

Shrestha, M., Zhang, Y., Upshur, M. A., Liu, P., Blair, S. L., Wang, H., Nizkorodov, S. A., Thomson, R. J., Martin, S. T., and Geiger, F. M.: On surface order and disorder of α-pinene-derived secondary organic material, J. Phys. Chem. A, 10.1021/jp510780e, 2014.

Zhang, Y., Sanchez, M. S., Douet, C., Wang, Y., Bateman, A. P., Gong, Z., Kuwata, M., Renbaum-Wolff, L., Sato, B. B., Liu, P. F., Bertram, A. K., Geiger, F. M., and Martin, S. T.: Changing shapes and implied viscosities of suspended submicron particles, Atmos. Chem. Phys., 15, 7819-7829, 10.5194/acp-15-7819-2015, 2015.

---

## Author Comment (AC1) · 25 Jun 2018

Dear co-editor and referees,

We appreciate all your detailed and valuable comments on our manuscript of "acp-2018-262". Please see the point-by-point response below and changes are marked blue in the revised manuscript.

**Referee #1**

This paper describes an observation of organosulfates (OSs) as well as nitroxyorganosulfates (NOSs) in Beijing over a summer period. A number of OSs and NOSs are detected by direct infusion electrospray mass spectrometry. In addition, several OSs are quantified by HPLC-ESI methods using authentic standards or surrogates. The measurements are very done and cover a meaningfully long period of time making it possible to qualitatively correlate the presence of OSs or NOSs with other environmental conditions, such as precursor concentrations, liquid water content, particle acidity etc. The paper should be published after some minor improvements described below.

*In my opinion, the introduction section does not explain the motivation of the study sufficiently well. The introduction sections states that OSs are important, were observed many times, can form by mechanisms that are affected by environmental conditions, and can now be in some cases quantified with authentic standards. However, it does not explain what this study is trying to accomplish and how it builds on all the previous field and lab work on OSs and NOSs. There is a clue about what authors want to accomplish in the sentence on lines 89-90, but this is not enough. Ideally, a testable hypothesis or a clear set of goals should be posed in the introduction.*
*Another issue I have with the introduction is that it motivates the study by saying that we cannot predict the amount of SOA correctly. The implication is that formation of rather involatile OSs and NOSs should help resolve this discrepancy. However, the mass concentrations of OSs reported here are rather small compared to the mass concentration of organic matter in particles. Since OSs and NOSs are minor species, it is better to motivate the study by our quest to understand the acid-catalyzed chemistry in particles, of which formation of OSs is an example, and the night time chemistry of $NO_3$ in urban environments, which appears to be partly responsible for the observed NOSs.*

**Response**: Thanks for your suggestions. The goals and motivation of this study were added or revised as suggested in lines 91-97, 98-105:

Lines 91-97: "Missing knowledge of formation mechanisms, the complexities of ambient aerosol composition and oxidation condition, and the lack of commercially available standards all hinder us from understanding the formation and fate of OSs in ambient atmosphere. Few field studies has been conducted in urban areas dominated by anthropogenic pollutants (e.g. $NO_x$, $SO_4^{2-}$). Observations are lacking to illustrate how severe anthropogenic pollutants could influence the OS formation under different

physical environmental conditions. This work reports a comprehensive characterization of particulate OSs in summertime Beijing, a location under the influence of both biogenic and severe anthropogenic sources. This study provides direct observational evidence for gaining insights into OS formation."

Lines 98-105:

"…HPLC-MS was then applied to quantify some OSs and NOS species in ambient aerosols using newly synthesized authentic standards and surrogate standards."

"Previously proposed formation pathways of OS or NOS (e.g. acid-catalyzed aqueous-phase chemistry, nighttime $NO_3$ chemistry) were considered, and the influence of different environment conditions or factors on the formation were comprehensively elaborated."

"It has been suggested that both aqueous-phase chemistry and nighttime $NO_3$ chemistry play important roles in the heavy haze of Beijing (Wu et al., 2018; Wang et al., 2017b; Wang et al., 2017a). Using OSs and NOSs as examples, this work illustrates SOA formation via acid-catalyzed aqueous-phase chemistry, nighttime $NO_3$ chemistry under the interaction between abundant anthropogenic pollutants and biogenic emissions."

*I found the discussion in sections 3.3 and 3.5 too qualitative. I realize that the data set mat be not long enough to make more definitive conclusions. Tied to this is Figure 4, the data in which are too scattered to make any reliable conclusions. I am not sure what the authors can do about it under the circumstances. Is Figure 4 the best way to look for correlation in the data? Have the authors tried correlating the amount of observed OSs to, for example, a product of sulfate, organic mass, and hydronium ion concentrations that might be expected to describe an acid-catalyzed reaction? I would encourage them to come up with more convincing ways of presenting the data then currently afforded by Figure 4.*

**Response**: The correlation analysis was added in Table S4, Fig. 8 and the main text (lines 331-333, 357-359, 433-434, 452-454, ).

Lines 331-333: The OS concentrations generally followed a similar trend to that of sulfate aerosols (Fig. 3). The total OS concentrations showed strong correlations with sulfate (r=0.67) or aerosol acidity (r=0.67), suggesting the driving role of acidic sulfate aerosols in the OS formation (Table S4).

Lines 357-359: Stronger correlations between OSs and sulfate (r=0.67) or aerosol acidity (r=0.67) compared with that between OSs and LWC (r=0.55) also suggest the importance of acid-catalyzed chemistry for OSs formation.

Lines 433-434: Both isoprene OSs and NOSs showed strong correlations with isoprene oxidation products (MVK+MACR) (Table S4).

Line 452-454: During daytime, the correlation of isoprene NOSs with $NO_2$ (r=0.74) was stronger than that with MVK+MACR (r=0.69) (Fig. 8).

Lines 457-459: During nighttime, a strong correlation between isoprene NOS and MVK+MACR (r=0.94) was observed, while the increase trend of isoprene NOSs as a function of $NO_2$ (r=0.53) was not so obvious and their correlation was lower (Fig. 8).

*Here are minor editorial comments.*

*L57-58: this is an unclear sentence, please revise*
    **Response**: It has been revised as "The sulfate esterification of alcohols could also be a pathway leading to OSs formation, while Minerath et al (2018) predicted that this mechanism was kinetically insignificant under ambient tropospheric conditions. However, this prediction was based on laboratory bulk solution-phase experiments and the applicability to the liquid-phase on particles suspended in the air is unconfirmed." (lines 58-61)

*L104: monoterpene -> monoterpenes*
    **Response**: Revised accordingly (line 114).

*L124: mass resolution -> mass resolving power*
    **Response**: Revised accordingly (line 136).

*L128: Please be more explicit about the allowed elements and constraints placed on the formulas*
    **Response**: The allowed elements and constraints placed on the formulas were described in lines 140-144: "Elements $^{12}C$, $^{1}H$, $^{16}O$, $^{14}N$, $^{32}S$ and $^{13}C$ were allowed in the molecular formula calculations. The H/C, O/C, N/C and S/C ratios were limited to 0.3- 3.0, 0- 3.0, 0- 0.5 and 0- 2.0. The assigned formulas were also restrained by the double bond equivalent values and the nitrogen rule for even electron ions. More details about the molecular formula assignment have been introduced in Wang et al. (2017c)."

*L145: OSs and NOS species -> OS and NOS species*
    **Response**: Revised accordingly (line 161).

*L149: Olson et al. (2011) (Olson et al., 2011) -> Olson et al. (2011). Similar corrections may be needed in other places in the manuscript, for example, on line 153*
    **Response**: Thanks. All similar cases have been corrected throughout the main text.

*L171: please clarify what are "compounds excluded from the above major compound categories". Would it be CHS? Or peaks that could not be assigned within the imposed constraints? It would also be useful to know what fraction of peaks was assigned.*
    **Response**: Related descriptions were added in lines 201-202: "'others' (e.g. CH, CHN, CHS, CHNS) refer to the compounds excluded from the above major compound categories" and line 197: "On average, 62% of the observed peaks in ESI negative mode are assigned with unambiguous molecular formulas."

*L202: OSs molecules -> OS molecules*
    **Response**: Revised accordingly (line 234).

*L202: monoterpene -> monoterpenes*
    **Response**: Revised accordingly (line 235).

*L203: molecules -> ions*
    **Response**: Revised accordingly (line 235).

*L205: relatively higher relative intensity -> higher relative intensity*

**Response**: Revised accordingly (line 237).

*L206: less -> fewer*

**Response**: Revised accordingly (line 240).

*L221: does the "total concentrations of quantified OSs" refer the average over all the samples?*

**Response**: Yes, it's the average over all the samples. This sentence was revised as "The average concentration of all the quantified OSs were 41.4 ng/m$^3$ during the campaign." (lines 256-257)

*L227: Where is "Centreville" located? (Unlike the other locations mentioned, it is not a country or state)*

**Response**: Centreville is located in Alabama, US. The study in Centreville was combined with other studies in Alabama, US to be clear.

*L268: monoterpene -> monoterpenes*

**Response**: Revised accordingly (line 315).

*L270: were -> was*

**Response**: Revised accordingly (line 317).

*L289 and elsewhere in this section: Secondary inorganic aerosols (SIAs) are defined as sulfate, nitrate and ammonium on line 289. However, the ratio of $SO_4^{(2-)}$ to SIAs above or below 0.5 is then used to separate conditions into sulfate and nitrate dominated regimes. The exact definition of this ratio is not clear. To be more precise the authors should define a molar ratio $SO_4^{(2-)}/[SO_4^{(2-)}+NO_3^{(-)}]$ and use it in their discussion instead of the vaguely defined ratio they are currently using.*

**Response**: The "$SO_4^{2-}$/SIAs ratios" was revised to "$SO_4^{2-}$/SIAs mass concentration ratios", to make the definition clear in section 3.3. SIAs (secondary inorganic aerosols) or SNA (sulfate, nitrate, and ammonium) have been commonly used to indicate the secondary inorganic aerosols (sulfate, nitrate, and ammonium) in previous studies (Chen et al., 2016; Wu et al., 2018; Zheng et al., 2016; Aksoyoglu et al., 2017; Choi et al., 2009; Alastuey et al., 2004; Jimenez-Guerrero et al., 2011; Huang et al., 2014). Each of the three secondary ions (i.e., sulfate, nitrate, and ammonium) is important in influencing aerosol liquid water content and acidity (Guo et al., 2015). Thus, we prefer to keep the definition of "SIAs" to indicate different aerosol compositions during the three episodes.

**References:**

Aksoyoglu, S., Ciarelli, G., El-Haddad, I., Baltensperger, U., and Prévôt, A. S. H.: Secondary inorganic aerosols in Europe: sources and the significant influence of biogenic VOC emissions, especially on ammonium nitrate, Atmos. Chem. Phys., 17, 7757-7773, 10.5194/acp-17-7757-2017, 2017.

Alastuey, A., Querol, X., Rodríguez, S., Plana, F., Lopez-Soler, A., Ruiz, C., and Mantilla, E.: Monitoring of atmospheric particulate matter around sources of secondary inorganic aerosol, Atmos. Environ., 38, 4979-4992, 10.1016/j.atmosenv.2004.06.026, 2004.

Chen, D., Liu, Z., Fast, J., and Ban, J.: Simulations of sulfate–nitrate–ammonium (SNA) aerosols during the extreme haze events over northern China in October 2014, Atmos. Chem. Phys., 16, 10707-10724,

10.5194/acp-16-10707-2016, 2016.

Choi, Y.-S., Park, R. J., and Ho, C.-H.: Estimates of ground-level aerosol mass concentrations using a chemical transport model with Moderate Resolution Imaging Spectroradiometer (MODIS) aerosol observations over East Asia, J. Geophys. Res., 114, 10.1029/2008jd011041, 2009.

Guo, H., Xu, L., Bougiatioti, A., Cerully, K. M., Capps, S. L., Hite, J. R., Carlton, A. G., Lee, S. H., Bergin, M. H., Ng, N. L., Nenes, A., and Weber, R. J.: Fine-particle water and pH in the southeastern United States, Atmos. Chem. Phys., 15, 5211-5228, 10.5194/acp-15-5211-2015, 2015.

Huang, R. J., Zhang, Y., Bozzetti, C., Ho, K. F., Cao, J. J., Han, Y., Daellenbach, K. R., Slowik, J. G., Platt, S. M., Canonaco, F., Zotter, P., Wolf, R., Pieber, S. M., Bruns, E. A., Crippa, M., Ciarelli, G., Piazzalunga, A., Schwikowski, M., Abbaszade, G., Schnelle-Kreis, J., Zimmermann, R., An, Z., Szidat, S., Baltensperger, U., El Haddad, I., and Prevot, A. S.: High secondary aerosol contribution to particulate pollution during haze events in China, Nature, 514, 218-222, 10.1038/nature13774, 2014.

Jimenez-Guerrero, P., Jose Gomez-Navarro, J., Jerez, S., Lorente-Plazas, R., Garcia-Valero, J. A., and Montavez, J. P.: Isolating the effects of climate change in the variation of secondary inorganic aerosols (SIA) in Europe for the 21st century (1991–2100), Atmos. Environ., 45, 1059-1063, 10.1016/j.atmosenv.2010.11.022, 2011.

Wu, Z. J., Wang, Y., Tan, T. Y., Zhu, Y. S., Li, M. R., Shang, D. J., Wang, H. C., Lu, K. D., Guo, S., Zeng, L. M., and Zhang, Y. H.: Aerosol Liquid Water Driven by Anthropogenic Inorganic Salts: Implying Its Key Role in Haze Formation over the North China Plain, Environmental Science & Technology Letters, 5, 160-166, 10.1021/acs.estlett.8b00021, 2018.

Zheng, J., Hu, M., Peng, J., Wu, Z., Kumar, P., Li, M., Wang, Y., and Guo, S.: Spatial distributions and chemical properties of $PM_{2.5}$ based on 21 field campaigns at 17 sites in China, Chemosphere, 159, 480-487, 10.1016/j.chemosphere.2016.06.032, 2016.

*L304-305: please fix grammar in this sentence*

**Response**: The sentence was revised as following:

Lines 354-356: "During the daytime of May 23, higher aerosol LWC was observed due to the rapid increase of nitrate, however, the aerosol acidity was lower as a result of the less contribution from sulfate."

*L345 and 352: the "nighttime formation" and "lower production" are not plausible explanations. The diurnal profile of monoterpenes may be peaking at night because they are removed more slowly at night and/or boundary layer height is changing. It should say "nighttime peak in concentrations" instead of "nighttime formation".*

**Response**: "nighttime formation" was changed to "elevated nighttime concentrations" (lines 399-400).

"lower production" was deleted in the revised version.

"The lower concentrations of monoterpene NOSs during the daytime could be attributed to the much lower production, as the monoterpene, $NO_x$ and $NO_x$/BVOCs ratios were much lower than those at night." was changed to "The lower concentrations of monoterpene NOSs during the daytime could be attributed to the lower monoterpene, $NO_x$ and $NO_x$/BVOCs ratios than those at night." (lines 409-410)

*Table 1: some of the OS structures are draw as ionized (deprotonated) and some are not. Since the formulas are given for the ionized forms, it would be good to draw the structures for the ionized forms as well for consistency*

**Response**: All the OS structures in Table 1 were changed to deprotonated forms.

---

## Author Comment (AC2) · 25 Jun 2018

Dear co-editor and referees,

We appreciate all your detailed and valuable comments on our manuscript of "acp-2018-262". Please see the point-by-point response below and changes are marked blue in the revised manuscript.

**Referee #2**

This study investigates the composition, abundance, and formation pathways of organosulfates during the day and night in the summer of Beijing under the influence of biogenic and anthropogenic emissions. Under various pollution episodes characterized by different aerosol composition and levels of gas-phase pollutants, the authors show that the formation of organosulfates can be largely influenced by the concentrations and relative contribution of sulfate, aerosol acidity and liquid water content. This work provides new field observation data to better understand the abundance and formation of organosulfates in the atmosphere. I support the publication of this paper after addressing the following questions.

*Major comments:*
*My major comment is the calculation and determination of the aerosol acidity and aerosol phase water content.*
*Line 161, "Aqueous phase $[H^+]$ and LWC were then calculated with the ISORROPIA-II thermodynamic model. ISORROPIA-II was operated in forward mode, assuming the particles are "metastable" (Hennigan et al., 2015; Weber et al., 2016; Guo et al., 2015). The input parameters included: ambient RH, temperature, particle phase inorganic species ($SO_4^{2-}$, $NO_3^-$, $Cl^-$, $NH_4^+$, $K^+$, $Na^+$, $Ca^{2+}$, $Ma^{2+}$) and gaseous $NH_3$."*

*First, the authors should provide justifications why ambient aerosols can be assumed to be "metastable" in this work. It is important to discuss the physical state of the ambient aerosols (e.g. solid, liquid or solid/liquid) during the field campaign. This information is important for analyzing and interpreting the data given aqueous aerosol-phase reactions have been proposed as one of the major formation pathways for the organosulfates in this work. Some major conclusions are drawn based on this assumption. A detailed explanation is needed.*

**Response**:

(1) Thanks for the suggestions. The detailed explanation for assuming "metastable" has now been added in Appendix S2.

The ISORROPIA-II thermodynamic model was run for metastable aerosols in this study. It has been suggested in previous studies that "metastable" state often showed better performance than the "stable" state solution, and was commonly applied in previous pH or LWC predictions (Bougiatioti et al., 2016; Guo et al., 2016, 2015, 2017; Weber et al., 2016; Liu et al., 2017a). The verification of thermodynamic prediction by ISORROPIA-II was assessed by comparing the predicted and measured gaseous $NH_3$ in this study (Fig. S2). Good agreement was reached between the predicted and measured

gaseous ammonia concentrations (slope=0.99, intercept= 1.8 μg/m$^3$, R$^2$= 0.97). The result suggested that the "metastable" assumptions are reasonable in this study.

The detailed explanation was added in Appendix S2 and the main text (lines 188-189).

Lines 188-189: "The thermodynamic calculations were validated by the good agreement between measured and predicted gaseous NH$_3$ (slope=0.99, R$^2$= 0.97) (see Appendix S2 for details)."

(2) We agree that phase state of ambient aerosols is important for discussing the aqueous aerosol-phase reactions. The ubiquitous existence of ambient metastable aerosols has been observed in previous studies (Rood et al., 1989). Liquid phase state of ambient aerosols has also been observed during haze episode in winter Beijing (Liu et al., 2017b). Unfortunately, we lack direct evidence to reveal the aerosol phase state in this study. Related discussion was also added in Appendix S2.

**Appendix S2 The validation of ISORROPIA-II thermodynamic model prediction**

The ISORROPIA-II thermodynamic model was run for metastable aerosols in this study. It has been suggested in previous studies that "metastable" state (only liquid phase) often showed better performance than the "stable" state (solid+ liquid) solution, and was commonly applied in previous pH or LWC predictions (Bougiatioti et al., 2016;Guo et al., 2015;Guo et al., 2016;Guo et al., 2017;Weber et al., 2016;Liu et al., 2017a). Though we lack direct evidence to reveal the physical state of ambient aerosols in this study, indirect evidence is provided to support that the assumption is reasonable. The verification of prediction by ISORROPIA-II was assessed by comparing the predicted and measured gaseous NH$_3$ in this study (Fig. S2) (Bougiatioti et al., 2016;Guo et al., 2015;Guo et al., 2016;Guo et al., 2017;Weber et al., 2016;Liu et al., 2017a). Good agreement was reached between predicted and measured gaseous ammonia concentrations (slope=0.99, intercept= 1.8 μg/m$^3$, R$^2$= 0.97). The result suggested that the "metastable" assumptions are reasonable in this study.

[Figure]

Figure S2 Comparison of predicted NH$_3$ (g) and measured NH$_3$ (g)

The ubiquitous existence of ambient metastable aerosols has been observed in previous studies (Rood et al., 1989). Based on our previous study in the winter of urban Beijing (Liu et al., 2017b), the rebound fraction of fine particles was ~0.8 at <20% RH, indicating a semisolid phase of particles. As the RH increased from 20 to 60%, the rebound fraction decreased from 0.8 to 0.2, suggesting transition from semisolid to liquid

phase state. The rebound fraction of particles was lower than 0.4 at >40% RH, indicating that the liquid phase as the major phase state of ambient aerosols. RH conditions of < 20% were quite limited during the campaign. RH was usually higher than 40% and could increase to higher than 60% at night. Thus, a nearly liquid phase was likely the major phase state of ambient aerosols at night in this study. It would be desirable to obtain direct observational evidence of aerosol phase state in future studies.

**References:**

Bougiatioti, A., Nikolaou, P., Stavroulas, I., Kouvarakis, G., Weber, R., Nenes, A., Kanakidou, M., and Mihalopoulos, N.: Particle water and pH in the eastern Mediterranean: source variability and implications for nutrient availability, Atmos. Chem. Phys., 16, 4579-4591, 10.5194/acp-16-4579-2016, 2016.

Guo, H., Xu, L., Bougiatioti, A., Cerully, K. M., Capps, S. L., Hite, J. R., Carlton, A. G., Lee, S. H., Bergin, M. H., Ng, N. L., Nenes, A., and Weber, R. J.: Fine-particle water and pH in the southeastern United States, Atmos. Chem. Phys., 15, 5211-5228, 10.5194/acp-15-5211-2015, 2015.

Guo, H., Sullivan, A. P., Campuzano-Jost, P., Schroder, J. C., Lopez-Hilfiker, F. D., Dibb, J. E., Jimenez, J. L., Thornton, J. A., Brown, S. S., Nenes, A., and Weber, R. J.: Particle pH and the Partitioning of Nitric Acid during Winter in the Northeastern United States, J. Geophys. Res., [Atmos.], 121, 10355-10376, 10.1002/2016JD025311, 2016.

Guo, H., Weber, R. J., and Nenes, A.: High levels of ammonia do not raise fine particle pH sufficiently to yield nitrogen oxide-dominated sulfate production, Scientific reports, 7, 12109, 10.1038/s41598-017-11704-0, 2017.

Liu, M., Song, Y., Zhou, T., Xu, Z., Yan, C., Zheng, M., Wu, Z., Hu, M., Wu, Y., and Zhu, T.: Fine particle pH during severe haze episodes in northern China, Geophys. Res. Lett., 44, 5213-5221, 10.1002/2017gl073210, 2017a.

Liu, Y., Wu, Z., Wang, Y., Xiao, Y., Gu, F., Zheng, J., Tan, T., Shang, D., Wu, Y., Zeng, L., Hu, M., Bateman, A. P., and Martin, S. T.: Submicrometer Particles Are in the Liquid State during Heavy Haze Episodes in the Urban Atmosphere of Beijing, China, Environmental Science & Technology Letters, 4, 427-432, 10.1021/acs.estlett.7b00352, 2017b.

Weber, R. J., Guo, H., Russell, A. G., and Nenes, A.: High aerosol acidity despite declining atmospheric sulfate concentrations over the past 15 years, Nature Geosci., 9, 282-285, 10.1038/ngeo2665, 2016.

*Second, given the organic compounds contribute significantly to the total aerosol mass in this work, do the organic compounds being considered when the aerosol acidity and aerosol water content are calculated using the aerosol thermodynamic model. If not, the authors should discuss how the organic compounds would affect the predictions of the aerosol acidity and aerosol water content. Would the findings or conclusions drawn from the data based on inorganic species only affect by the presence of organic compounds? The authors should discuss how they determine the ambient RH and temperature used for their model simulations.*

**Response**: New discussion has been added in main text (lines 189-194, 177-179) and Fig. S3 to address this point.

Lines 189-194: "The contribution of organics to LWC was not considered in this study. Our previous study in Beijing has suggested that LWC associated with organic species was insignificant (<6%), compared to that of secondary inorganic aerosols (Wu et al., 2018) (see Fig. S3 for the comparison between LWC with or without water associated with

organic compounds). Previous study also suggested that the predicted aerosol acidity or pH without consideration of organic water could also be sufficient for discussing aqueous SOA chemistry in this study, due to the minor effect on aerosol pH (0.15-0.23) (Guo et al., 2015)."

Thus, the conclusions drawn from the data of inorganic species would not be affected by the presence of organic compounds, due to their minor effects on aerosol water or acidity.

[Figure]

Figure S3 Comparison between aerosol liquid water content with or without water associated with organic compounds. LWCi and LWCo represent the water contributed by inorganic compounds and organic compounds, respectively. The data is from Fig. S1 in Wu et al. (2018) based on the measurement in Beijing (Wu et al., 2018).

Lines 177-179: "Meteorological parameters, including relative humidity (RH), temperature, wind direction and wind speed (WS) were continuously monitored by a weather station (Met one Instrument Inc.) during the campaign."

*Minor comments:*

*Line 147 "OSs and NOSs were quantified using authentic standards or surrogates with similar molecular structures (Table 1). Lactic acid sulfate (LAS) and glycolic acid sulfate (GAS) were prepared according to Olson et al. (2011) (Olson et al., 2011). Four monoterpene derived OSs were synthesized according to Wang et al. (2017) (Wang et al., 2017d)." What is the purity of these standards used in this work? What is the recovery or extraction efficiency of these standards?*

**Response**: The purity and recovery of OS standards were added in lines 164-168 and table S1.

Lines 164-168: "Lactic acid sulfate (LAS) and glycolic acid sulfate (GAS) were prepared according to Olson et al. (2011). The purity of LAS and GAS are 8% and 15%, determined by 1H NMR analysis using dicholoracetic acid as an internal standard, and the recovery are 89.5% and 94.9%, respectively. Four monoterpene derived OS standards were synthesized and the details are given in Wang et al. (2017). The purity of the four monoterpene OS standards are higher than 99% and the recovery are 80.5%-93.5% (Table S1)."

Table S1 The purity and recovery of organosulfate standards in this study

| Organosulfate | Purity (%) | Recovery (%) |
|---|---|---|
| lactic acid sulfate | 15% | |
| glycolic acid sulfate | 8% | |
| α-pinene OS | >99% | 80.5% |
| β-pinene OS | >99% | 93.5% |
| limonene OS | >99% | 85.4% |
| limonaketone OS | >99% | 82.5% |

*Line 172, "The OC content in each sample for Orbitrap MS analysis was kept roughly constant to minimize variation arising from matrix ion suppression." Please elaborate how to achieve this goal.*

**Response**: The variation of different sample matrix would influence the responses of different species (Furey et al., 2013). A previous study suggested that the extent of ion suppression showed good linearity with the concentrations of urine extracts (Chen et al., 2015).

A set of experiments were also conducted to evaluate the influence of sample matrix and ion suppression in this study. With constant OC loading, the variation of ion suppression extent arising from different chemical compositions was lower than 40% in this study. Overall, the extent of ion suppression was proposed to be comparable for samples with similar OC concentrations in this study, though the variation of ion suppression caused by different sample compositions cannot be eliminated. The related description has been added in the supplement (Appendix S1).

**Appendix S1 The influence of ion suppression on Orbitrap MS analysis**

The overall molecular composition of S-containing organic species was measured using ESI-Orbitrap MS analysis. Sample matrix would influence the MS responses of different species, which cannot be eliminated (Furey et al., 2013). A previous study showed the extent of ion suppression was in good linearity with the concentrations/dilution factors of urine extracts (Chen et al., 2015). We conducted a set of experiments to evaluate the influence of sample matrix on MS response for OSs. A field blank sample, a clean sample and a polluted sample were extracted following the same procedures described in section 2.2. The sample collected during the nighttime of May 24 and 30 were selected to represent the clean sample and the polluted sample, respectively. Sample extracts were dried and re-dissolved either in acetonitrile/water (1:1) solvent or solvent containing 0.1 ppm α-pinene OS. The filter portion size and solvent volume were adjusted to yield solution containing ~100 μg or 200 μg OC/mL solvent for the clean sample and polluted sample. The OC concentrations are referred to as the OC loading before the SPE clean-up procedure. Only two concentration levels were examined due to the sample limitation. The intensity (signal-to-noise ratio, s/n) of α-pinene OS (0.1 ppm) in the three different sample matrixes were obtained by deducting the intensity of m/z=249.0802 in the same sample diluted by solvent without α-pinene OS (0.1 ppm) addition. The intensity in each sample was normalized by the ion injection time to make the intensities comparable (Kuang et al., 2016).

The intensity of 0.1 ppm α-pinene OS was the highest in the matrix of field blank extract and the lowest in the matrix of polluted sample extract. The extent of suppression ranged from 20% in the matrix of 100 μg OC from the clean sample to 62% in the matrix of 200 μg OC from the polluted sample (Fig. S1). It was clear that the extent of suppression increased with the OC content of the matrix, from 20% in 100 μg OC matrix to 32% in 200 μg OC matrix for the clean sample and from 45% in 100 μg OC matrix to 62% in 200 μg OC matrix for the polluted sample. The relative standard deviation (RSD) of α-pinene OS arising from different OC loadings (100 μg and 200 μg OC/mL solution) were 26% and 12% in polluted and clean samples, respectively. This result confirmed the benefit of adjusting OC content to a uniform level before Orbitrap MS analysis in minimizing the impact of matrix ion suppression. We note that when the sample was diluted to 100μg OC/mL solvent, the intensity of α-pinene OS in the clean sample was comparable to that in the field blank sample. This indicated that the ion suppression would be insignificant with less than 100 μg OC/mL solution. However, this level of dilution may limit the identification of species present at low concentrations due to too much dilution.

It is also apparent that chemical composition of the OC matrix also played a role in ion suppression. The RSD of α-pinene OS arising from different chemical composition (clean sample and polluted sample) were 40% and 27% in samples containing 200 μg and 100 μg OC/mL solution, respectively, which could represent the biggest differences of ion suppression arising from chemical composition. This source of difference in ion suppression could not be controlled with the infusion injection mode.

[Figure]

Figure S1 The intensity of α-pinene OS (0.1 ppm) in different sample matrix (blank sample, clean sample, polluted sample) with different OC loadings. The error bars were derived from three repeat injections of the same sample.

*Line 177, "What's more, the S-containing compounds contributed more to the higher MW formulas than CHO ($O_1$-$O_{10}$) or CHON ($O_1$-$O_{11}$) compounds (Fig. 1), due to the existence of more O (CHOS: $O_1$-$O_{12}$, CHONS: $O_1$-$O_{14}$) atoms and heteroatoms (S, N) in the molecules. They may play more important roles in the increase of SOA mass concentrations during pollution episodes." Given the concentration of quantified organosulfates is small, the formation of organosulfates is not likely explained the increase of SOA mass. Please*

*elaborate this point.*

**Response**:

During pollution episodes, the number and intensity of S-containing compounds (CHOS and CHONS) increased obviously (Fig. S4). Considering the higher MW of CHOS and CHONS than those of CHO or CHON, the S-containing compounds may play more important roles in the increase of SOA mass concentrations. The second sentence was revised to make it clearer: "The increasing trend of S-containing organics (Fig S4), with larger MW than those of CHO or CHON, may play important roles in the increase of SOA mass concentrations during pollution episodes." (lines 210-211)

This conclusion was drawn based on the Orbitrap data shown in Fig. S4, rather than the OS concentrations quantified by HPLC-MS. A total of 351 OSs and 181 NOSs formulas were identified during the whole campaign, while only 13 selected OS and NOS species were quantified due to the lack of more standards. Though the quantified OS concentrations were low, the total concentrations of OSs in ambient atmosphere should be higher than those quantified in this study.

*Line 233, "carboxylic acids mainly form via aqueous phase oxidation in cloud or particle water, including both biogenic and anthropogenic sources (Charbouillot et al., 2012;Chebbi and Carlier, 1996). The relatively higher level of hydroxycarboxylic acid sulfate could be attributed to the favorable interaction between sulfate aerosols and carboxylic acids or other precursors in summertime Beijing, while the precursors and mechanisms remain unclear." The authors should provide more information or field data to support this argument. For example, what are the concentrations of these carboxylic acids if these acids have been measured in this field campaign?*

**Response**:

The concentration of oxalic acid, usually the most abundant dicarboxylic acid in the atmosphere, was added in lines 273-277: "Oxalic acid is usually the most abundant dicarboxylic acid in the atmosphere (Guo et al., 2010; Narukawa et al., 2003). The average concentration of oxalic acid in fine particles was 0.22 $\mu g/m^3$, which was at a relatively high concentration level when comparing with those reported in previous studies (0.02-0.32$\mu g/m^3$) (Agarwal et al., 2010; Bikkina et al., 2017; Boreddy et al., 2017; Deshmukh et al., 2017; Kawamura et al., 2010; Narukawa et al., 2003)."

*Line 260 "unclear. The concentration of isoprene NOSs ($C_5H_{10}NO_9S^-$) was lower than that of individual isoprene OSs. Strong inter-correlations were observed between isoprene OSs and NOSs (Table S2), suggesting their similar formation pathways." Please elaborate what are the formation pathways.*

**Response**:

The formation pathways were elaborated in lines 307-309: "Strong inter-correlations were observed between isoprene OSs and NOSs (Table S4), suggesting their similar formation pathways via acid-catalyzed epoxide chemistry (Worton et al., 2013)."

*For the 3.3 OS formation via acid-catalyzed aqueous phase chemistry, I understand the authors focus on understanding how the quantified organosulfates form under different pollution episodes. Do the authors observe the same results for the "unquantified" organosulfates (or other detected organosulfates) as well?*

**Response**:

The acid-catalyzed aqueous phase chemistry is suggested to be an important pathway for OS formation, based on the analysis of quantified OSs in section 3.3. The total intensity of OSs also followed similar temporal variation to that of quantified OSs (Fig. S4, Fig. 3(f)). This observation indicated acid-catalyzed aqueous phase chemistry could be an important or major pathway for OS formation, however, it would be too speculative to comment on the formation pathway of the unquantified OSs based on the direct injection measurement. We will learn about the formation pathway of more OS species through expanding available OS and NOS standards or combining with other techniques (e.g. isotopic analysis) in our future studies.

*Line 294 "Moreover, the higher aerosol LWC encountered during these periods would favor the uptake of gas-phase reactants into particle phase, due to the decrease of viscosity and increase of diffusivity within the particles (Shiraiwa et al., 2011)." These descriptions would be too qualitative. What would be the viscosity of the ambient aerosols expected in this work? How would the aerosol viscosity change with the aerosol composition and environmental conditions (e.g. ambient temperature and RH) in this work?*

**Response**:

We lack direct observation evidence to be more quantitative about the viscosity of ambient aerosols in this study. It has been suggested that the aerosol viscosity would decrease as the increase of ambient RH or temperature (Shiraiwa et al., 2011). Aerosol viscosity is also expected to decrease as the increase of secondary inorganic aerosols (SIAs, sulfate, nitrate, and ammonium), because hygroscopic SIAs would favor the increase of aerosol liquid water (Wu et al., 2018).

A nearly liquid phase (viscosity nearly or lower than $10^2$) was expected based on our previous measurement of aerosol rebound factors in the winter Beijing (Liu et al., 2017b). As the RH increased from 20 to 60%, the rebound fraction decreased from 0.8 to 0.2, meaning that the particles undergo the transition from semisolid to liquid phase state. The rebound fraction of particles was lower than 0.4 at >40% RH, indicating that the liquid phase as the major phase state of ambient aerosols. The conditions with RH< 20% were quite limited during the campaign. The RH was usually higher than 40% and could increase to higher than 60% at night. What's more, the temperature in this study was higher than that in winter, which would decrease the viscosity of ambient aerosols. Thus, a nearly liquid phase (viscosity nearly or lower than $10^2$) was expected at night, while semisolid phase with higher viscosity may occur during daytime when RH was very low. It would be desirable to observe the viscosity of ambient aerosol directly in future studies.

*Line 321, in the section, the authors elucidate the major factors influencing the formation of quantified organosulfates and their interrelations with SIA compositions. Do the authors also observe the same results for the unquantified organosulfates (or other detected organosulfates)?*

**Response**:

The influencing factors for OS formation were drawn based on the analysis of quantified OSs. The total intensity of OSs also followed similar temporal variation to that of quantified OSs (Fig. S4, Fig. 3(f)). This result indicated that the conclusion could be

applicable to most of the unquantified OSs. However, we feel it may be overly speculative to draw the conclusion only based on the infusion injection analysis using Orbitrap MS.

---

## Author Comment (AC3) · 25 Jun 2018

Dear co-editor and referees,

We appreciate all your detailed and valuable comments on our manuscript of "acp-2018-262". Please see the point-by-point response below and changes are marked blue in the revised manuscript.

**Referee #3**

This manuscript describes measurements of organosulfates and related compounds in Beijing during May-June 2016. Emphasis is placed on understanding the factors that influence organosulfate and nitrooxyorganosulfate formation, particularly during three pollution episodes. The manuscript concludes that sulfate, liquid water content, and acidity are important factors in their formation.

The measurements appear to be carefully conducted and well-described. However, there are some shortcomings in the presentation of the data that should be addressed prior to publication.

*Major comments:*

*1) The discussion of the trends in organosulfates and co-located measurements in section 3.3 is limited to qualitative descriptions. Correlation analysis (like that conducted between observed organosulfates, Table S2) should be extended to include co-located measurements of sulfate, nitrate, ammonium, liquid water content, aerosol acidity [H⁺], VOC precursors, oxidants, etc. to provide quantitative support for the associations (or lack thereof) that are discussed in this section.*

**Response**:

The correlation analysis in Table S4 is now extended to include sulfate, nitrate, ammonium, liquid water content, aerosol acidity [H$^+$], VOC precursors and oxidants. The related descriptions or analysis have also been added in the main text (lines 332-333, 357-359).

Lines 332-333: "The total OS concentrations showed strong correlations with sulfate (r=0.67) or aerosol acidity (r=0.67), suggesting the driving role of acidic sulfate aerosols in the OS formation (Table S4)."

Lines 357-359: "Stronger correlations between OSs and sulfate (r=0.67) or aerosol acidity (r=0.67) compared with that between OSs and LWC (r=0.55) also suggest the importance of acid-catalyzed chemistry for OSs formation."

*2) Correlation analysis should also be conducted and presented to support the discussion in section 3.5.*

**Response**:

The correlations between isoprene OSs/NOSs and the co-located measurements are now in Table S4. The correlations between isoprene NOSs and MVK+MACR or NO$_2$ are shown in Fig. 8. The related descriptions or analysis have also been added in the main text (lines

433-434, 452-454, 457-459).

Lines 433-434: "Both isoprene OSs and NOSs showed strong correlations with isoprene oxidation products (MVK+MACR) (Table S4)."

Lines 452-454: "During daytime, the correlation of isoprene NOSs with $NO_2$ (r=0.74) was higher than that with MVK+MACR (r=0.69) (Fig. 8). When MVK+MACR was higher than 0.7 ppb, the NOS concentrations did not increase further with MVK+MACR."

Lines 457-459: "During nighttime, a strong correlation between isoprene NOSs and MVK+MACR (r=0.94) was observed, while the increase trend of isoprene NOSs as a function of $NO_2$ (r=0.53) was not so obvious and their correlation was lower (Fig. 8)."

*3) The phrase "representative organosulfates" is used in several instances (line 22, 218, 411) although the authors do not indicate what these species represent. Rather than using this vague language, the authors should more explicit in describing why the selected compounds were quantified and semi-quantified.*

**Response**:

These species were selected as their precursors or formation mechanisms have been proposed in previous chamber studies and their formation represent the anthropogenic-biogenic interactions. The proposed mechanisms could be applied in the field observation. "representative" was deleted in the revised version (lines 22, 250, 471-472). New text given below is added to improve the clarity on this point.

Lines 251-255: "The quantified species could usually be formed via the interaction between biogenic precursors (e.g. isoprene, monoterpene) and anthropogenic pollutants (e.g. $SO_4^{2-}$, $NO_x$), which have been reported in previous chamber studies (Surratt et al., 2007; Surratt et al., 2008; Surratt et al., 2010). A total of ten OSs and three NOS species were quantified in this study and their concentrations are listed in Table 1."

*4) The discussion in the paragraph beginning at line 209 implies that the only difference between the three air pollution episodes was their inorganic ion content (which affected aerosol acidity and liquid water content). Do back trajectories, VOC concentrations, and other co-located measurements support this? If not, how could variations in other atmospheric conditions explain the organosulfate observations?*

**Response**:

The back trajectories, VOC and oxidant concentrations during each episode are now included added in Table S5. Description for the related analysis has been added in the main text (lines 335-336, 345-347, 359-360, 406-408, 440-442).

Lines 335-336: "The back trajectories, average concentrations of VOC precursors and oxidants during each episode are also shown in Table S5."

Lines 345-347: "Moreover, the oxidant levels, indicated by $O_x$ ($NO_2+O_3$) in this study (Herndon et al., 2008), were much higher than the other two episodes, which favored the formation of VOC oxidation products (e.g. MVK+MACR) (Table S5). This is another reason for higher OSs concentration level during episode III."

Lines 359-360: "The back trajectories during episode I were different from those during episode II or III (Table S5), which could be one reason for different conditions (e.g. SIA composition) during episode I."

Lines 406-408: "The highest nighttime concentration of $C_{10}H_{16}NO_7S^-$ was recorded on May

27 during episode II (Fig. 5). Besides the high $NO_2$ concentration (>20 ppb), the high monoterpene level was another primary reason for the elevated concentration of monoterpene NOSs (Table S5)."

Lines 440-442: "The highest concentrations of isoprene OSs and NOSs were observed during the nighttime of May 30 during episode III (Fig. 7), with high sulfate, MVK+MACR, aerosol acidity and LWC (Fig. 3, Table S5)."

*5) The overall concentration of organosulfates observed in Beijing seems to be very low (~150 ng/m$^3$). In encourage the authors to discuss this observation and include it in their comparison to prior studies.*

**Response**:

The key species and total quantified OS concentrations in this and prior studies are summarized and compared in Table S3. Related description or analysis was also added in the main text.

The low OS concentrations in Beijing compared with that in southeast US was mainly attributed to the low concentrations of isoprene OSs, especially $C_5H_{11}O_7S^-$ formed via $HO_2$ channel under low-$NO_x$ conditions (Table S3). The related discussions were added in lines 287-298. The reasons include: 1) The isoprene concentration in southeastern US (1.9 ppb) (Xu et al., 2015) is much higher than that observed during our campaign (297 pptv). 2) The IEPOX formation could be suppressed by the high-$NO_x$ conditions in Beijing (Zhang et al., 2017; Hu et al., 2015). 3) The RH in Beijing was lower than that in southeast US (Xu et al., 2015), which possibly led to an increase of aerosol viscosity and decrease of diffusivity within the particles, resulting in lower OS formation (Shiraiwa et al., 2011). 4) The OM-coated particle structures observed in Beijing could reduce the reactive uptake of isoprene oxidation products (Li et al., 2016; Zhang et al., 2018; Riva et al., 2016a), which may be another possible reason for lower isoprene OSs in this study. 5) Lactic acid sulfate was employed as a surrogate standard to quantify isoprene OSs, which may also be one possible reason for low isoprene OSs in this study.

Lines 287-298: "We used lactic acid sulfate as a surrogate standard to quantify isoprene OSs on the basis of their similar structures and retention times (Table 1). The isoprene concentration in southeastern US (1.9 ppb) (Xu et al., 2015) was much higher than that observed during our campaign (297 pptv). Besides the lower VOC precursors and measurement uncertainty, the lower isoprene OSs in this study could be attributed to different atmospheric conditions in Beijing from those in southeastern US. The IEPOX formation under low-NOx conditions ($HO_2$ channel), usually with higher yields than the oxidation products under high-$NO_x$ conditions (NO/$NO_2$) (Worton et al., 2013), could be suppressed under the high-$NO_x$ conditions (see section 3.4 for the high-$NO_x$ conditions) in Beijing (Zhang et al., 2017; Hu et al., 2015). The RH in Beijing was lower than that in southeast US (Xu et al., 2015), which possibly led to an increase of aerosol viscosity and a decrease of diffusivity within the particles, resulting in lower OS formation (Shiraiwa et al., 2011). Moreover, the OM-coated particle structures observed in Beijing could reduce the reactive uptake of isoprene oxidation products (Li et al., 2016; Zhang et al., 2018; Riva et al., 2016a), which may be another possible reason for lower isoprene OSs in this study."

*6) A table comparing key species and total organosulfate concentrations across this and prior studies would be a useful addition to the supplement to support the comparison of data.*

**Response**:

A summery table (Table S3) is now added in the supplement to compare the key species and total OSs concentrations in prior studies and this study.

*7) In two places (line 117 and 173) the authors indicate that the organic carbon concentration was held constant across samples analyzed by Orbitrap, in order to decrease ion suppression. The authors should provide a reference to support this statement and/or evidence to support that ion suppression did not occur.*

**Response**: The variation of different sample matrix would influence the responses of different species (Furey et al., 2013). A previous study suggested that the extent of ion suppression showed good linearity with the concentrations of urine extracts (Chen et al., 2015).

A set of experiments were also conducted to evaluate the influence of sample matrix and ion suppression in this study. With constant OC loading, the variation of ion suppression extent arising from different chemical compositions was lower than 40% in this study. Overall, the extent of ion suppression was proposed to be comparable for samples with similar OC concentrations in this study, though the variation of ion suppression caused by different sample composition cannot be eliminated. The related description has been added in the supplement (Appendix S1).

**Appendix S1 The influence of ion suppression on Orbitrap MS analysis**

The overall molecular composition of S-containing organic species was measured using ESI-Orbitrap MS analysis. Sample matrix would influence the MS responses of different species, which cannot be eliminated (Furey et al., 2013). A previous study showed the extent of ion suppression was in good linearity with the concentrations/dilution factors of urine extracts (Chen et al., 2015). We conducted a set of experiments to evaluate the influence of sample matrix on MS response for OSs. A field blank sample, a clean sample and a polluted sample were extracted following the same procedures described in section 2.2. The sample collected during the nighttime of May 24 and 30 were selected to represent the clean sample and the polluted sample, respectively. Sample extracts were dried and re-dissolved either in acetonitrile/water (1:1) solvent or solvent containing 0.1 ppm α-pinene OS. The filter portion size and solvent volume were adjusted to yield solution containing ~100 μg or 200 μg OC/mL solvent for the clean sample and polluted sample. The OC concentrations are referred to as the OC loading before the SPE clean-up procedure. Only two concentration levels were examined due to the sample limitation. The intensity (signal-to-noise ratio, s/n) of α-pinene OS (0.1 ppm) in the three different sample matrixes were obtained by deducting the intensity of m/z=249.0802 in the same sample diluted by solvent without α-pinene OS (0.1 ppm) addition. The intensity in each sample was normalized by the ion injection time to make the intensities comparable (Kuang et al., 2016).

The intensity of 0.1 ppm α-pinene OS was the highest in the matrix of field blank extract and the lowest in the matrix of polluted sample extract. The extent of suppression ranged from 20% in the matrix of 100 μg OC from the clean sample to 62% in the matrix

of 200 μg OC from the polluted sample (Fig. S1). It was clear that the extent of suppression increased with the OC content of the matrix, from 20% in 100 μg OC matrix to 32% in 200 μg OC matrix for the clean sample and from 45% in 100 μg OC matrix to 62% in 200 μg OC matrix for the polluted sample. The relative standard deviation (RSD) of α-pinene OS arising from different OC loadings (100 μg and 200 μg OC/mL solution) were 26% and 12% in polluted and clean samples, respectively. This result confirmed the benefit of adjusting OC content to a uniform level before Orbitrap MS analysis in minimizing the impact of matrix ion suppression. We note that when the sample was diluted to 100μg OC/mL solvent, the intensity of α-pinene OS in the clean sample was comparable to that in the field blank sample. This indicated that the ion suppression would be insignificant with less than 100 μg OC/mL solution. However, this level of dilution may limit the identification of species present at low concentrations due to too much dilution.

It is also apparent that chemical composition of the OC matrix also played a role in ion suppression. The RSD of α-pinene OS arising from different chemical composition (clean sample and polluted sample) were 40% and 27% in samples containing 200 μg and 100 μg OC/mL solution, respectively, which could represent the biggest differences of ion suppression arising from chemical composition. This source of difference in ion suppression could not be controlled with the infusion injection mode.

[Figure]

Figure S1 The intensity of α-pinene OS (0.1 ppm) in different sample matrix (blank sample, clean sample, polluted sample) with different OC loadings. The error bars were derived from three repeat injections of the same sample.

*8) The SPE method described at line 118 indicates that select compounds are removed by the SPE process. However, there is no mention until line 207-215 what effect the SPE has on organosulfates. I suggest making a note at line 120 indicating that further discussion of the impact of this clean-up procedure on organosulfates is discussed in section 3.1.*

**Response**:
A note was added in lines 130-132.
Line 130-132: "Some selected OS species with low MW would also be removed by the SPE clean-up procedure, which will be discussed in section 3.1."

*9) Line 150 – please explain how semi-quantification is achieved for a surrogate standard that appears to be comprised of a mixture of compounds (e.g., "alpha-pinene OS").*

**Response**:

This point was explained in lines 171-173: "For the molecule with isomers, quantification was performed by summing up the peak areas of the isomers, treated as one species (e.g., monoterpene NOSs with [M-H]⁻ at m/z 294 were treated as one NOS species)."

*10) The discussion at lines 198-202 implicates long-chain alkanes and diesel/biodiesel emissions as the source of several organosulfates. Can the authors please comment on the (un)certainty of these assignments and the possibility that they may derive from monoterpenes (given the similarities in the molecular formulas to the monoterpene derived organosulfates mentioned later in the same paragraph)?*

**Response**:

The comment on the uncertainty when assigning OSs sources was added in lines 231-232: "Many OSs previously designated as biogenic origins were also found in the anthropogenic sources (Blair et al., 2017), which may raise uncertainty when assigning OS sources in field observation studies."

*11) Line 225-227, please include the city, state, and country for each of the measurement sides discussed. Centreville and summertime Alabama are presented as though they are different locations, when they are one in the same.*

**Response**:

The studies reported in Alabama were combined together. The city, state, and country were included in the summery table (Table S3), and the state or country information were included in the main text.

*12) I encourage the authors to consider their use of significant figures in reporting their data. Many organosulfate concentrations are listed to four significant figures, while their contributions to organic carbon have only one. The former seems to be too many (considering measurement uncertainties and use of surrogate standards) and the latter seems to be not enough.*

**Response**:

Revised. The OS concentrations are now presented to show three significant figures while the contributions of OS to OM are shown with two significant figures.

*13) In Table S2, please label which compounds are "isoprene OS" that are mentioned at line 238.*

**Response**: "isoprene OSs" is labelled in Table S4.

*14) In several places, the wording should be adjusted so as to better reflect that many species were semi-quantified and absolute concentrations remain unknown. At line 244: "The concentration of quantified isoprene OS…" At line 271 "…were the second most abundant signals among the observed species…"*

**Response**: They were revised accordingly (lines 284, 318).

*15) In Figure 1, please write out the dates "24 May night" and "30 May night" rather than "0524N" and "0530N".*

**Response**: Revised accordingly.

*16) In the figure 2 caption, please point the reader to the specific section where the information about the missing water-soluble OS can be found (3.1)*

    **Response**: Do you mean the figure 1 caption? It was revised as "…, details are described in section 3.1".

*17) In the Figure 2 caption, please explain that these plots only include the select species quantified or semi-quantified by LCMS.*

    **Response**:

    The caption was revised as "The relative contribution of different OS and NOS species. Only the selected species (semi-)quantified by HPLC-MS are included in this figure."

*18) There is a lot going on in Figure 4. Can this be simplified? Or perhaps broken into multiple graphs that do not have so much overlap? Also, because there is so much going on, adding the key findings / takeaway messages from the graphs to the caption would help the reader.*

    **Response**:

    Figure 4 was broken into four graphs and takeaway message was added: "When sulfate dominated the accumulation of secondary inorganic aerosols ($SO_4^{2-}$/SIAs> 0.5), both aerosol LWC and acidity (pH<2.8) increased and OS formation was obviously promoted. In comparison, the acid-catalyzed OS formation was limited by lower aerosol acidity under nitrate-dominant conditions."

*19) Figure S1 – delete 2016 from the date on the x-axis (as this takes up unnecessary room). It would be helpful to designate what is daytime and nighttime in this figures as is done in many of the other figures.*

    **Response**:

    Revised as suggested. '2016' was deleted from the date on x-axis. The gray background was added to denote the nighttime and white background was used to denote the daytime.

*20) Table S1 would benefit from organization by m/z so that the table can be easily navigated by other researchers who are likely to look up the data in this way.*

    **Response**: Revised as suggested. The formulas in Table S2 have been organized by m/z in the revised version.

**Technical/editorial comments**

*21) Line 20: "the majority"*

    **Response**: Revised accordingly (line 20).

*22) Line 22: "mass spectrometry was employed"*

    **Response**: Revised accordingly (line 22).

*23) Define LWC at line 27*

    **Response**: LWC was defined in line 27.

*24) Define SIA species at line 28*

**Response**: SIAs was defined in line 28.

25) *"0.02%" of OA at line 46 seems too small, is this reasonable?*
   **Response**: Thanks very much for your careful reading. The percent "0.02%" should was "2%". It has been corrected (line 47).

26) *Line 109: "flow rates were"*
   **Response**: Revised accordingly (line 119).

27) *Line 164: $Mg^{2+}$ (rather than $Ma^{2+}$)*
   **Response**: Thanks. It was corrected (line 188).

28) *Line 172: do not need to say "percent" in either instance, since it is earlier in the sentence.*
   **Response**: Revised accordingly (line 202-203).

29) *Line 283: "favorable for OS formation"*
   **Response**: Revised accordingly (line 330).

30) *Hettiyadura et al. (2015) propose a mechanism for the formation of the isoprene organosulfate with m/z 211 that is consistent with the hypothesis presented by Surratt et al. (2008).*
   **Response**: Thanks for the reminding.

31) *Line 343: "times larger than daytime"*
   **Response**: Revised accordingly (line 397).

32) *Line 349: "levels at night…"*
   **Response**: Revised accordingly (line 403).

33) *Line 351: "was in excess and no longer the limiting factor in NOS formation.*
   **Response**: Revised accordingly (line 405-406).

34) *Line 380: "NOSs form via"*
   **Response**: Revised accordingly (line 439).

35) *Line 382: "formation of isoprene OSs or NOSs, epoxides first form…"*
   **Response**: Revised accordingly (line 442).

36) *Line 393: "increase further with MVK+MACR."*
   **Response**: Revised accordingly (line 453-454).

37) *Line 419: "OS concentrations"*
   **Response**: Revised accordingly (line 479).

38) *Line 430: "$NO_2$ levels at night…"*
   **Response**: Revised accordingly (line 490).

---

## Author Comment (AC4) · 25 Jun 2018

Dear co-editor and referees,

We appreciate all your detailed and valuable comments on our manuscript of "acp-2018-262". Please see the point-by-point response below and changes are marked blue in the revised manuscript.

**Referee #4**

This ACPD article characterizes the amount of organosulfates (OSs) and nitrooxy organosulfates (NOS) through ESI-Orbitrap and HPLC-MS during a field campaign near Beijing, China. By using standards or surrogates, the manuscript breaks down organosulfates into isoprene-derived OSs and monoterpene derived OSs, and shows isoprene-derived OSs dominated the total OSs.

By analyzing inorganic aerosol composition, acidity, and liquid water content, the author concludes that due to acid catalyzed chemistry, the production of isoprene-derived OSs was strongly correlated with the acidity of the particles, which was governed by sulfate percentage in secondary inorganic aerosols.

The monoterpene-derived NOSs and isoprene-derived NOSs were measured mainly by the HPLC-MS and their concentrations were used to correlate with the ambient NOx concentration. Monoterpene NOSs were greatly enhanced during night time due to high NOx concentration.

As written in the manuscript, through measurements of OSs and NOSs, this study describes the interaction between biogenic emission and anthropogenic pollutants. The data of this kind are valuable and fits into the scopes of ACP. Overall, the manuscript is sound and after addressing the following issues, it is suitable to be published on ACP.

**Comments**

*Line 145: This part was not very clear. How did the author obtain the monoterpene NOSs for quantification? Was it synthesized or commercially available? Please illustrate in detail.*

**Response**:

Monoterpene NOSs were quantified using α-pinene OSs or limonaketone OS as surrogates due to their similar structures shown in Table 1. The monoterpene OSs were synthesized and the details are reported in Wang et al. (2017). These have been illustrated in lines 166-167, 170-171.

Line 166-167: "Four monoterpene derived OS standards were synthesized and the details are given in Wang et al. (2017)."

Lines 170-171: "α-pinene OS and limonaketone OS were respectively used to quantify monoterpene NOSs $C_{10}H_{16}NO_7S^-$ and $C_9H_{14}NO_8S^-$ due to the similar carbon structures (Table 1)."

*Line 239: The manuscript describes the strong correlation between GAS, LAS, and HAS with isoprene OSs. Then the author concludes that "isoprene or its oxidized products as potential*

*precursors of GAS, LAS and HAS."* The logic here is flawed. Isoprene OSs are correlated with sulfate concentration. Therefore, it is very much likely that GAS, LAS, and HAS are just correlated with sulfate concentration. The correlation of GAS with isoprene OSs does not mean isoprene oxidation products may be precursors of GAS, LAS, and HAS. I suggest changing this part to: *"They also showed strong correlations with isoprene OSs (Table S2), suggesting GAS, LAS, and HAS can be potential tracers for organosulfates."*

> **Response**:
>
> The sentence has been revised as below to improve the clarity:
>
> Lines 278-280: "They also showed strong correlations with isoprene oxidation products (MVK+MACR) and isoprene OSs (Table S4), suggesting isoprene oxidized products as potential precursors of GAS, LAS and HAS."
>
> It has been suggested that GAS, LAS and HAS could form via isoprene oxidation in the presence of acidic sulfate (Fu et al., 2008; Carlton et al., 2009; Riva et al., 2016a; Surratt et al., 2008; Schindelka et al., 2013). (as described in lines 280-283) Thus, we indicate here that isoprene oxidized products could be potential precursors of GAS, LAS and HAS.

*Line 250: The author raised a very interesting point here. The Southeast U.S. has an isoprene OSs concentration of 165 ng $cm^{-3}$ (Rattanavaraha et al., 2016). The isoprene concentration in Beijing is only 5 times lower than Southeast U.S, but the isoprene-derived OS is 10 times lower. The average RH in Beijing is also lower than Southeast U.S. Maybe the author should provide this evidence to further support the statement that organic coatings and their phase can play an important role.*

> **Response**: Thanks for the suggestions. This was added in lines 294-295: "The RH in Beijing was lower than that in southeast US (Xu et al., 2015), which possibly led to an increase of aerosol viscosity and a decrease of diffusivity within the particles, resulting in lower OS formation (Shiraiwa et al., 2011)."

*There are a few important references that I would suggest adding to the manuscript:*

*Line 69: I suggest adding Shrestha et al., 2014; Zhang et al., 2015 to provide more evidence for how RH and LWC affect aerosol viscosity.*

> **Response**: The references were added in lines 73.

*Line 66: I suggest adding Riedel et al., 2015 to show acid-catalyzed reactive uptake reactions in forming isoprene-derived OA.*

> **Response**: The reference was added in lines 70.

*Line 250: I suggest adding Riva et al., 2016 to show the effects of pre-existing organic loading on isoprene-derived SOA formation. Riva et al. was the first to show the effect of OA on isoprene SOA formation, and Zhang et al. 2018 was the first to quantify such effects.*

> **Response**: Thanks for the reminding. The reference (Riva et al., 2016a) was added in line 297.

*Line 164: Oxford comma is suggested here.*

> **Response**: Revised as suggested (line 188).

*Line 371-372: Are there any evidence to show that isoprene NOSs are formed via similar pathways (or multiphase reactions) as isoprene OSs? To my knowledge there is limited*

*experimental evidence to show the isoprene NOSs are formed through multiphase reactions.*
*The author should provide more evidence to support the argument.*

**Response**:

The sentence was revised to be accurate: "Formation of the isoprene NOSs are supposed to have similar limiting factors to those affecting isoprene OSs". (lines 428-429)

$NO_3$-initiated oxidation was proposed as the limiting step in the formation of monoterpene NOSs, supported by the observation of nighttime enhancement under high-$NO_x$ conditions. Acid-catalyzed chemistry was proposed to be a limiting step in the formation of isoprene OSs. We lack direct evidence to discern whether -$ONO_2$ group in isoprene NOSs was added in gas-phase or multiphase-phase reaction processes. However, we note isoprene NOSs showed similar temporal variation and good correlations with sulfate, aerosol acidity and isoprene OSs. This led us to propose that acid-catalyzed step, rather than $NO_3$-initiated (-$ONO_2$ addition) step was the limiting step in isoprene NOSs formation. In other words, the isoprene NOSs did not appear to form via $NO_3$-initiated oxidation as monoterpene NOSs.

---

## Author Response (AR2)

Dear Dr. Jason Surratt,

Thank you very much for your technical suggestions on our manuscript of "acp-2018-262". Please see the point-by-point responses below and changes are marked blue in the revised manuscript.

Thank you for considering all 4 reviewer comments. After careful re-reading of the revised manuscript, I feel you have clearly addressed these comments. However, upon reading the newest version, I found several minor edits/technical corrections that must be corrected. Once you correct these, I'll gladly accept this for final publication in ACP.

Minor Edits/Technical Corrections:

*1.) Page 1, Lines 16-17: change to: "we chemically characterized OSs and nitrooxy OSs (NOSs) formed under the influence of....."*
  **Response**: Changed accordingly (line 16-17).

*2.) Page 1, Line 18: change "The" to "A ultrahigh-resolution"*
  **Response**: Changed accordingly (line 18).

*3.) Page 1, Line 21: delete the word "the' before "high"*
  **Response**: Revised accordingly (line 21).

*4.) Page 2, Line 42: Insert the word "the" between "in" and "ambient"*
  **Response**: Revised accordingly (line 42).

*5.) Page 2, Line 49:*

*Change "Many chamber experiments studied try to reveal" to "Many prior chamber experiments revealed the precursors....."*
  **Response**: Changed accordingly (line 49).

*6.) Pages 2-3, Lines 49-51: Please add Surratt et al. (2007, ES&T) to this citation.*
  **Response**: The reference is now added (line 49).

*7.) Page 3, Line 51: Change ", which remain unclear" to "however, the atmospheric relevance of these remains unclear."*
  **Response**: Changed accordingly (line 51).

*8.) Page 5, Line 101: Please change "environment" to "environmental"*
  **Response**: Changed accordingly (line 101).

*9.) Page 5, Line 118: Should the authors consider adding ", respectively." after "..., China)" ?*
  **Response**: Revised accordingly (line 118).

*10.) Page 6, Line 129: Can the authors clarify how much of the isoprene-derived OSs were not captured by your direct infusion method due to the use of SPE? I know from personal experience that when we analyzed $PM_{2.5}$ samples collected from the southeastern U.S. and applied SPE, we did not detect isoprene-derived OSs (Gao et al., 2016, JGR). As a result, in*

*Surratt et al. (2017, ES&T), we found that there were indeed present when using LC/MS and not employing an SPE pretreatment step.*

**Response**: Our results are consistent with your previous work (Gao et al., 2006; Surratt et al., 2007). Isoprene-derived OSs were not detected in SPE-treated samples, while they were detected in samples without undergoing the SPE pretreatment procedure. We now have added the following text to page 6 to clarify the impact of the SPE treatment on OSs detected.

Lines 132-136:

[revised manuscript text omitted]